# Growth-dependent signals drive an increase in early G1 cyclin concentration to link cell cycle entry with cell growth

Robert A Sommer, Jerry T DeWitt, Raymond Tan, Douglas R Kellogg*

Department of Molecular, Cell, and Developmental Biology, University of California, Santa Cruz, Santa Cruz, United States

**Abstract** Entry into the cell cycle occurs only when sufficient growth has occurred. In budding yeast, the cyclin Cln3 is thought to initiate cell cycle entry by inactivating a transcriptional repressor called Whi5. Growth-dependent changes in the concentrations of Cln3 or Whi5 have been proposed to link cell cycle entry to cell growth. However, there are conflicting reports regarding the behavior and roles of Cln3 and Whi5. Here, we found no evidence that changes in the concentration of Whi5 play a major role in controlling cell cycle entry. Rather, the data suggest that cell growth triggers cell cycle entry by driving an increase in the concentration of Cln3. We further found that accumulation of Cln3 is dependent upon homologs of mammalian SGK kinases that control cell growth and size. Together, the data are consistent with models in which Cln3 is a crucial link between cell growth and the cell cycle.

## Introduction

Cell cycle progression is subservient to cell growth. Thus, key cell cycle transitions occur only when sufficient growth has occurred. A particularly important point at which cell growth controls cell cycle progression is at the end of G1 phase, when cells decide whether to commit to cell cycle entry. The key molecular event that marks cell cycle entry is transcription of late G1 phase cyclins, which bind and activate cyclin-dependent kinases (CDKs) to initiate cell cycle events. Transcription of the late G1 cyclins is initiated only when sufficient growth has occurred.

A key question concerns how growth triggers cell cycle entry. Most models suggest that growth-dependent changes in the number or concentration of key cell cycle regulatory proteins trigger cell cycle entry. If the number of molecules of a protein increases at the same rate as growth, the number of molecules per cell will increase but the concentration of the protein will stay constant. In this case, an increase in the number of protein molecules per cell could be sufficient to trigger cell cycle entry if the activity of the protein is titrated against an activity that does not scale with growth. Alternatively, cell cycle entry could require changes in the concentration of cell cycle regulatory proteins. In this case, if the number of molecules per cell increases faster than growth, the concentration and activity of the protein should also increase. Conversely, if the number of protein molecules stays constant as growth occurs the concentration of the molecule will decrease. It remains unclear whether changes in number or concentration of cell cycle regulators trigger cell cycle entry. In addition, it remains possible that changes in the catalytic rate of an enzyme, such as a kinase or phosphatase, play an important role.

In budding yeast, key regulators of the cell cycle that are thought to link cell cycle entry to cell growth were first identified by genetic analysis. An early G1 cyclin called Cln3 was found to be a dose-dependent regulator of cell cycle entry and cell size (*Cross, 1988*; *Nash et al., 1988*). Overexpression of *CLN3* causes premature cell cycle entry at a reduced cell size, while loss of Cln3 causes delayed cell cycle entry at a large cell size. A poorly understood protein called Bck2 plays redundant roles with

*For correspondence:
dkellogg@ucsc.edu

Competing interest: The authors declare that no competing interests exist.

Cln3 (*Epstein and Cross, 1994*). Loss of Cln3 or Bck2 causes delayed cell cycle entry, whereas loss of both causes a failure in cell cycle entry.

The Cln3/Cdk1 complex triggers cell cycle entry by activating the SBF transcription factor, which drives transcription of late G1 phase cyclins, as well as hundreds of additional genes required for subsequent cell cycle events. The Cln3/Cdk1 complex is thought to activate SBF by directly phosphorylating and inactivating Whi5, a transcriptional repressor that keeps SBF inactive prior to cell cycle entry (*Jorgensen et al., 2002*; *Costanzo et al., 2004*; *de Bruin et al., 2004*). Loss of Whi5 causes premature cell cycle entry at a reduced cell size, which suggests that Whi5 plays an important role in the mechanisms that link cell cycle entry to cell growth (*Jorgensen et al., 2002*).

Early studies suggested a model in which accumulation of Cln3 protein drives cell cycle entry (reviewed in *Jorgensen and Tyers, 2004b*; *Turner et al., 2012*). Cln3 protein is rapidly turned over, which should make Cln3 levels highly sensitive to translation rate (*Tyers et al., 1992*). If translation rate increases with cell size, the number of Cln3 protein molecules in the cell should also increase with cell size, which could lead to cell cycle entry. Difficulties in detecting Cln3 protein initially made it difficult to test the prediction that Cln3 protein levels increase with cell growth during G1 phase. More recent studies found that levels of Cln3 increase gradually during G1 phase, reaching a peak around the time of bud emergence, consistent with the idea that gradually rising levels of Cln3 trigger cell cycle entry (*Zapata et al., 2014*). Another recent study used an in vivo reporter to measure the rate of Cln3 translation and found that the rate increases faster than the rate of growth in G1 phase, which suggests that an increase in the concentration of Cln3 could trigger cell cycle entry (*Litsios et al., 2019*). In contrast, another study that utilized a mutant stabilized overexpressed version of Cln3 concluded that the concentration of Cln3 does not change during growth (*Schmoller et al., 2015*). Overall, the relationship between Cln3 accumulation and cell growth remains poorly understood. It is unclear whether Cln3 provides a simple readout of overall translation rate or whether the rate of Cln3 accumulation is influenced by other growth-dependent signals. It is also unclear how regulation of Cln3 transcription, translation, and turnover influences accumulation of Cln3 during cell growth. Finally, it remains unknown whether or how a threshold level of Cln3 triggers cell cycle entry.

Another study suggested that dilution of Whi5 by cell growth is the critical event that links cell cycle entry to cell growth (*Schmoller et al., 2015*). In this study, it was found that the concentration of Cln3 remains constant during growth because the number of Cln3 molecules per cell increases in a manner that is commensurate with growth. In contrast, the concentration of Whi5 was found to decrease because no new Whi5 is produced during G1 phase. The model postulates that once Whi5 concentration drops below a threshold, the activity of Cln3 becomes sufficient to inactivate Whi5, thereby triggering cell cycle entry. A limitation of this model is that cells undergo little growth in G1 phase (*Ferrezuelo et al., 2012*; *Leitao and Kellogg, 2017*; *Litsios et al., 2019*), which means that Whi5 concentration is reduced by less than 50%  and by as little as 10–20%. It is unclear how such small changes in Whi5 concentration could be translated into an effective cell size control decision. Another limitation is that the effects of Whi5 protein concentration on cell cycle entry were analyzed only in *bck2Δ* cells, and only under nutrient poor conditions. Finally, the behavior of Cln3 protein was analyzed using a mutant version of Cln3 that lacks amino acid sequences that target it for proteolytic destruction, which results in overexpression and loss of cell cycle-dependent changes in protein levels.

Another limitation of current models for size control in G1 phase is that they fail to explain proportional relationships between cell size and growth rate that hold across all orders of life. Yeast cells growing slowly in poor nutrients are nearly twofold smaller than cells growing rapidly in rich nutrients (*Johnston et al., 1977*). Moreover, a proportional relationship between cell size and growth rate holds even when comparing yeast cells growing at different rates under identical nutrient conditions (*Ferrezuelo et al., 2012*; *Leitao and Kellogg, 2017*). For example, cells in a population of wild-type yeast cells in early G1 phase show a threefold variance in growth rate. The slow-growing cells within the population undergo cell cycle entry at a reduced cell size compared to rapidly growing cells. Thus, it appears that signals linked to nutrients and growth rate modulate the threshold amount of growth required for cell cycle progression. Classic experiments in fission yeast suggest that the threshold amount of growth required for cell cycle progression is readjusted within minutes of a shift to new nutrient conditions (*Fantes and Nurse, 1977*). A model that could explain the link between cell size and growth rate is that global signals that set growth rate also set the threshold amount of growth

required for cell cycle progression (*Lucena et al., 2018*). However, the mechanisms by which nutrient-dependent signals modulate growth thresholds remain largely unknown.

Distinguishing models will require a clear understanding of how changing concentrations of Cln3 and Whi5 influence cell cycle entry, and how the concentrations of Cln3 and Whi5 vary during growth and in response to changes in growth rate. However, there are conflicting reports on the behaviors of Cln3 and Whi5 during growth in G1 phase (*Tyers et al., 1993*; *Zapata et al., 2014*; *Liu et al., 2015*; *Schmoller et al., 2015*; *Lucena et al., 2018*; *Dorsey et al., 2018*; *Blank et al., 2018*; *Litsios et al., 2019*; *Barber et al., 2020*; *Black et al., 2020*). Most previous studies used quantitative fluorescence microscopy to analyze the behavior of tagged proteins in living cells. Differences in fluorescent tags, nutrient and imaging conditions, potential imaging artifacts, as well as the use of mutant proteins or mutant genetic backgrounds could explain differences in results. In addition, the low abundance of Cln3 has made it difficult to measure levels of wild-type Cln3 by fluorescence microscopy.

Here, we used quantitative western blotting in synchronized cells to carry out a comprehensive analysis of the function and regulation of Cln3 and Whi5 during growth in G1 phase. A simple model for nutrient modulation of cell size could be that poor nutrients cause a reduction in Whi5 protein levels, which would reduce the amount of growth required to dilute Whi5 below the threshold required for cell cycle entry. However, little is known about the relationship between Cln3 and Whi5 protein levels during growth in G1 phase in differing nutrient conditions. We therefore investigated how levels of the Whi5 and Cln3 proteins are modulated during growth in G1, and how nutrient availability modulates levels of both proteins. We also investigated the mechanisms that control accumulation of Cln3 during growth in G1 phase.

## Results

### Cells growing in poor carbon undergo cell cycle entry at a dramatically lower ratio of Cln3 to Whi5

Previous studies found that levels of Cln3 mRNA and protein are reduced in asynchronous cells growing under poor nutrient conditions (*Gallego et al., 1997*; *Parviz and Heideman, 1998*; *Hall et al., 1998*; *Blank et al., 2018*). However, more recent work discovered that Cln3 is also synthesized in mitosis (*Landry et al., 2012*; *Zapata et al., 2014*). Therefore, analysis of Cln3 in asynchronous cells could not define the behavior of Cln3 during growth in G1 phase. Here, we used synchronized cells to characterize the dynamics of Cln3 and Whi5 proteins during growth in G1 phase in rich versus poor carbon.

Cells were grown in a poor carbon source (YP media containing 2% glycerol and 2% ethanol) and centrifugal elutriation was used to isolate small newborn daughter cells in early G1 phase. The isolated cells were then resuspended in YP media containing rich carbon (2% dextrose) or poor carbon to initiate growth in G1 phase. An advantage of this approach is that the cells growing in rich versus poor carbon were previously grown under identical conditions and each culture contains identical numbers of cells, which allows comparison of levels of Cln3 and Whi5 between conditions. Another advantage is that the cells shifted to rich carbon undergo a prolonged interval of growth in G1 phase to reach the increased size that is characteristic of cells growing in rich carbon, which provides a longer interval for examining the behavior of Cln3 and Whi5.

Median cell size was determined with a Coulter Channelyzer and plotted as a function of time (*Figure 1A*, *Figure 1—figure supplement 1A*). Since bud emergence provides a marker of cell cycle entry, the percentage of cells with buds was plotted as a function of time (*Figure 1B*). Finally, a plot of the percentage of cells with buds versus cell size provided a measure of cell size at cell cycle entry (*Figure 1C*). As expected, the cells transferred to rich carbon grew faster and spent more time undergoing growth in G1 phase compared to cells in poor carbon. The cells in rich carbon also entered the cell cycle at a larger size than the cells in poor carbon. The cells that remained in poor carbon grew in size by only 24% before bud emergence (defined as the time at which 20% of cells were budded). In contrast, the cells shifted from poor to rich carbon increased in size by 75%.

The behaviors of Whi5 and Cln3 were assayed by western blot in the same samples. A constant number of cells was collected for each time point so that western blot signals provide a measure of protein copy number per cell. Whi5 was detected with a 3XHA tag, whereas Cln3 was detected with a 6XHA tag. Cln3 tagged with 3XHA could not be detected, which necessitated use of the

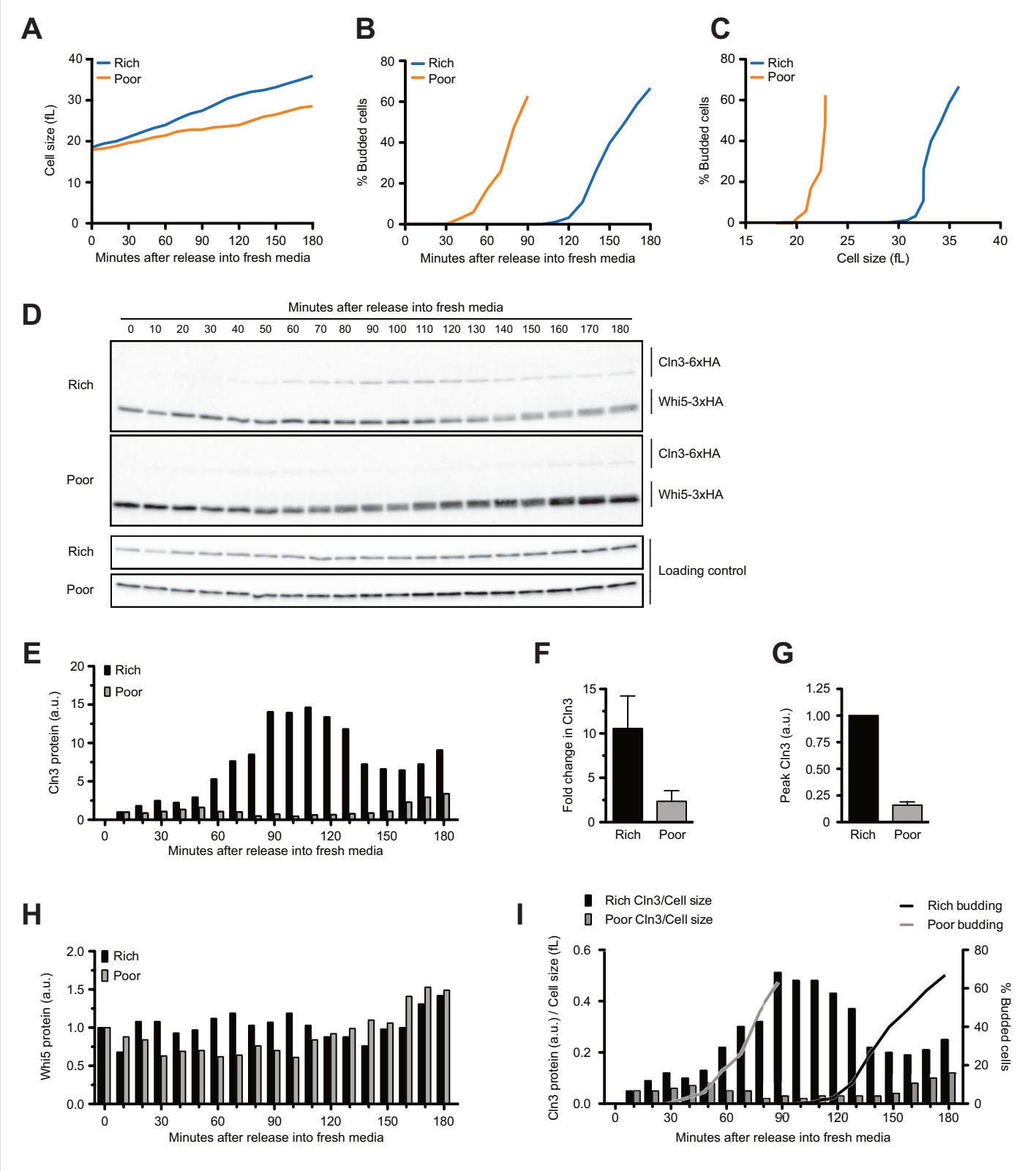

**Figure 1.** Dynamics of Cln3 and Whi5 proteins during G1 phase in cells growing in rich or poor carbon. Wild-type cells were grown to mid-log phase in poor carbon (YPG/E), and small unbudded cells were isolated by centrifugal elutriation. Cells were released into either rich carbon (YPD) or poor carbon (YPG/E) at 25°C , and samples were taken at 10 min intervals. All data in the figure are from the same biological replicate. (**A**) Median cell size was measured using a Coulter counter at 10 min intervals and plotted as a function of time. (**B**) The percentage of budded cells as a function of time.

*Figure 1 continued on next page*

*Figure 1 continued*

(**C**) The percentage of budded cells as a function of median cell size at each time point. (**D**) The behavior of Cln3-6XHA and Whi5-3XHA was analyzed by western blot on the same blot for each carbon source. An anti-Nap1 antibody was used as a loading control. (**E**) Quantification of Cln3-6XHA protein levels from (**D**) as a function of time. At each time point, the relative Cln3 signal was calculated as a ratio to the signal of the 10 min time point. (**F**) The fold change in peak Cln3 protein signal was calculated as a ratio over the signal from the 10 min time point. The data represent the average of three biological replicates. The error bars represent SEM. (**G**) The difference in peak Cln3 between rich and poor carbon was calculated by first normalizing the peak Cln3 signal to the loading control in each carbon source, and then comparing the signal between carbon sources. The error bars represent SEM. (**H**) Quantification of Whi5-3XHA from western blots in (**D**). The change in relative protein abundance over time was measured by calculating the ratio over the zero time point for each time point. (**I**) The relative change in Cln3 concentration over time was calculated by taking the ratio of the Cln3 signal over cell size at each time point. The bud emergence data from panel (**B**) are included for reference. The experiment shown in this figure was repeated for three biological replicates, which gave similar results. In panels (**E**), (**H**), and (**I**), the Cln3 and Whi5 signals were not normalized to a loading control because the loading control signal increases with growth.

The online version of this article includes the following figure supplement(s) for figure 1:

**Source data 1.** Source data for *Figure 1D*.

**Source data 2.** Source data for *Figure 1D*.

**Figure supplement 1.** Dynamics of Cln3 and Whi5 proteins during G1 phase in cells growing in rich or poor carbon.

**Figure supplement 1—source data 1.** Source data for *Figure 1—figure supplement 1C*.

**Figure supplement 2.** Dynamics of Cln3 and Whi5 during G1 phase in cells grown continuously in complete synthetic media containing dextrose.

**Figure supplement 2—source data 1.** Source data for *Figure 1—figure supplement 2C*.

**Figure supplement 2—source data 2.** Source data for *Figure 1—figure supplement 2C*.

---

6XHA tag. Tagging both Cln3 and Whi5 with 6XHA caused the proteins to co-migrate on SDS-PAGE, which prevented use of identical tags. The use of HA tags on both proteins allowed detection on the same western blot so that relative behaviors of Cln3 and Whi5 during growth could be directly compared. However, the use of tags with different detection sensitivities precluded comparison of absolute protein levels between Cln3-6XHA and Whi5-3XHA. Nevertheless, the fact that Whi5-3XHA is easily detectable, whereas Cln3-3XHA is undetectable, indicates that levels of Whi5 far exceed those of Cln3. Cln3-6XHA caused a slight increase in cell volume in rich carbon and had little effect on cell size in poor carbon (*Figure 1—figure supplement 1B*). A previous study found Cln3 tagged with protein A also causes a slight increase in cell volume but fully rescues loss of *CLN3* function in genetic backgrounds that require *CLN3* function for viability (*Cross et al., 2002*). Previous studies found that fluorescently tagged versions of Whi5 do not cause a decrease in cell size, which suggests that tagged versions of Whi5 are fully functional. Here, we found that Whi5-3XHA cells were slightly smaller than wild-type cells (*Figure 1—figure supplement 1B*). Whi5 tagged with a 13Myc tag did not cause a decrease in cell size, which suggests that the HA tag has a mild effect on Whi5 function. Whi5-3XHA and Whi5-13Myc showed the same behavior in elutriated cells undergoing growth in G1 phase. Use of Whi5-3XHA in these experiments allowed us to directly compare the behavior of Cln3-6XHA and Whi5-3XHA in the same samples and on the same western blots. Western blots comparing Cln3 and Whi5 in rich versus poor carbon were carried out simultaneously under identical conditions to allow direct comparison of protein levels between carbon sources. Here, we use 'concentration' to refer to the number of molecules per unit of cell volume and the more general term 'protein levels' to refer to the number of molecules in the cell.

In both conditions, Cln3 was not detectable at the 0 min time point (*Figure 1D*, see *Figure 1—figure supplement 1C* for a longer exposure). Cln3 was first detected at 10 min after elutriation and levels of Cln3 increased gradually during growth in both rich and poor carbon. Cln3 levels peaked slightly before the first buds could be detected in both conditions, consistent with a previous study that found that the rate of Cln3 translation peaks before Whi5 exits the nucleus (*Litsios et al., 2019*). Cln3 levels were dramatically lower in poor carbon. Quantification of the Cln3 signal relative to the 10 min time point in multiple biological replicates indicated that Cln3 protein levels increased approximately 10-fold before bud emergence in rich carbon and 2.5-fold in poor carbon (*Figure 1E and F*). Quantification of peak Cln3 levels in rich and poor carbon suggests that cells in poor carbon initiate cell cycle entry with approximately eightfold lower levels of Cln3 than cells in rich carbon (*Figure 1G*). Note that this population-level analysis could miss transient bursts in Cln3 translation rate that were

previously observed using a translation reporter in single cells (*Litsios et al., 2019*). Thus, accumulation of Cln3 protein in single cells may not be as gradual as indicated in the population-level analysis.

We did not detect differences in Whi5 protein levels between the two carbon sources. In *Figure 1D*, a slightly longer exposure was used for the western blot from cells in poor carbon so that Cln3 could be detected, but quantification of Whi5 levels relative to a loading control showed no difference between rich and poor carbon. To test whether Whi5 levels change during G1 phase, we calculated a ratio of the Whi5 signal at each time point over the Whi5 signal at the 0 min time point (*Figure 1H*). We used the same approach to calculate the fold change in Whi5 levels between the zero time point and bud emergence in multiple biological replicates (*Figure 1—figure supplement 1D*). Both plots suggest that levels of the Whi5 protein do not change substantially before bud emergence in rich or poor carbon, consistent with several previous studies that used microscopy to analyze fluorescently tagged versions of Whi5 (*Schmoller et al., 2015*; *Dorsey et al., 2018*; *Litsios et al., 2019*; *Black et al., 2020*).

A fraction of Whi5 shifted to lower electrophoretic mobility forms at the time of bud emergence, which likely corresponds to the activity of a previously described positive feedback loop in which the late G1 cyclins Cln1 and Cln2 activate Cdk1 to phosphorylate and inactivate Whi5 (*Cross and Tinkelenberg, 1991*; *Costanzo et al., 2004*). Note that the Whi5 phosphorylation is first detected 20–30 min after Cln3 protein reaches peak levels, consistent with phosphorylation being an indirect consequence of accumulation of Cln3.

We also attempted to assay Cln3 and Whi5 protein levels during G1 phase in cells grown continuously in rich medium (YP+ dextrose). However, we found that it was not possible to isolate a uniform population of small unbudded cells because very little growth occurs in G1 phase in rich medium, which means that newborn cells are nearly the same size as mother cells (*Leitao and Kellogg, 2017*). This, combined with normal variation in cell size observed in wild-type yeast, means that elutriation yields a mix of unbudded and budded cells. Previous studies also encountered this issue (*Futcher, 1999*; *Litsios et al., 2019*). As an alternative, we grew cells in complete synthetic medium (CSM) containing 2% dextrose. Under these conditions, cells grow slowly and produce small newborn cells because CSM is limiting for nutrients other than dextrose. We isolated small unbudded cells and released them into the same medium. We found that Cln3 protein levels increased threefold, while Whi5 protein levels did not change substantially (*Figure 1—figure supplement 2*), similar to the results obtained in YP medium containing poor carbon.

Together, the data show that Cln3 protein levels are correlated with the extent of growth in volume in G1 phase in both rich and poor carbon, reaching a peak near the time of bud emergence. In addition, overall Cln3 protein levels in G1 phase are correlated with the growth rate set by carbon source. Thus, Cln3 levels are high in rich carbon and low in poor carbon. In contrast, Whi5 protein levels are not modulated by carbon source. As a result, cells growing in poor carbon enter the cell cycle at a much lower ratio of Cln3 to Whi5 protein compared to cells in rich carbon.

The large increase in Cln3 protein levels in cells shifted from rich to poor carbon likely reflects a rapid resetting of the threshold amount of Cln3 required for cell cycle entry. In this case, the cells shifted to rich carbon would require a higher threshold of Cln3 for cell cycle entry, which would require more growth to accumulate sufficient Cln3 protein to drive cell cycle entry. The fact that there is no difference in Whi5 protein levels between rich and poor carbon suggests that Whi5 protein levels have little to do with setting threshold amount of Cln3 required for cell cycle entry.

## The concentration of Cln3 increases during growth in G1 phase

A previous study utilized quantitative fluorescence microscopy to analyze accumulation of a mutant stabilized overexpressed version of Cln3 and concluded that the concentration of Cln3 does not change during growth in complete synthetic media containing poor carbon (*Schmoller et al., 2015*). To estimate changes in wild-type Cln3 concentration before bud emergence, we divided the Cln3 signal from western blots by cell volume at each time point (*Figure 1I*). Using this approach, we quantified the increase in Cln3 concentration between the 10 min time point and the time of bud emergence for multiple biological replicates (*Figure 1—figure supplement 1E*). This showed that the concentration of Cln3 increased sevenfold prior to bud emergence in rich carbon and twofold in poor carbon. A rise in concentration of Cln3 protein during G1 phase was also detected in a previous study that used targeted proteomics to analyze the concentration of untagged wild-type Cln3 (*Litsios et al., 2019*).

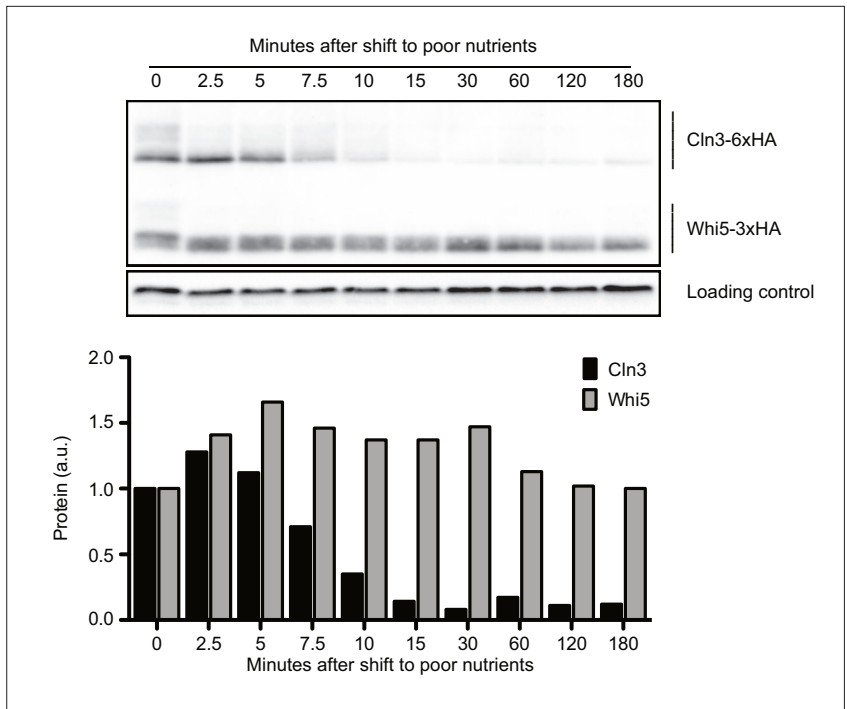

**Figure 2.** Cln3 protein levels respond rapidly to changes in nutrient availability. Rapidly growing cells in rich carbon (YPD) were shifted to poor carbon (YPG/E) at 25°C , and the behavior of Cln3-6XHA and Whi5-3XHA was analyzed by western blot on the same blot. The levels of Cln3 and Whi5 protein were quantified relative to the 0 min time point. An anti-Nap1 antibody was used as a loading control. To account for any differences in total protein after shifting to poor carbon, Cln3 and Whi5 signals were calculated as a ratio to that of the loading control, which was quantified relative to the 0 min time point.

The online version of this article includes the following source data for figure 2:

**Source data 1.** Source data for *Figure 2C*.

We carried out a similar analysis for Whi5 and found that Whi5 concentration decreased by approximately 30%  in rich carbon and 20%  in poor carbon, consistent with the fact that Whi5 protein levels do not change substantially and there is only a small increase in cell size in G1 phase (*Figure 1—figure supplement 1F*). Our results are consistent with several previous studies that observed a relatively small increase in cell volume in G1 phase (*Ferrezuelo et al., 2012*; *Leitao and Kellogg, 2017*) and little or no change in Whi5 concentration (*Dorsey et al., 2018*; *Litsios et al., 2019*; *Black et al., 2020*). A previous study found that a small fraction of cells growing in complete synthetic media containing poor carbon show a 30–40% decrease in Whi5 concentration, but most cells showed a decrease of 20–30% or less (*Schmoller et al., 2015*). Population-level analysis of Cln3 and Whi5 protein levels by western blotting could miss changes in Whi5 or Cln3 concentration driven by changes in localization to specific subcellular compartments.

## Cln3 protein levels respond rapidly to changes in nutrient availability

In both budding yeast and fission yeast, cells rapidly readjust the threshold amount of growth required for cell cycle progression when shifted to nutrient conditions that support different growth rates (*Fantes and Nurse, 1977*; *Johnston et al., 1979*; *Kief and Warner, 1981*; *Lucena et al., 2018*). In fission yeast, the threshold amount of growth appears to be readjusted within minutes. Therefore, cell cycle regulators that link cell cycle progression to cell growth should show a similar rapid response to changes in nutrients.

A previous study found that Cln3 protein levels decrease within 30 min of a shift from rich nitrogen medium to medium that completely lacks nitrogen, but did not test shorter time points or the effects of carbon source (*Gallego et al., 1997*). We therefore analyzed the behavior of Cln3 and Whi5 following a shift from rich to poor carbon. Cln3 underwent rapid changes in abundance when asynchronous cells

were shifted from rich to poor carbon (*Figure 2*). The amount of Cln3 protein transiently increased within 5 min and then rapidly decreased to become nearly undetectable. In contrast, Whi5 levels remained constant (*Figure 2*). Whi5 phosphorylation was lost during the time course, consistent with previous studies showing that cells in poor nutrients spend more time in early G1 phase, which is when Whi5 is found in a dephosphorylated state (*Hartwell and Unger, 1977*). These observations show that levels of Cln3, but not Whi5, respond rapidly to nutrient-dependent signals that modulate cell growth and size.

## Increased dosage of *WHI5* does not cause large effects on cell size

The discovery that cells in rich or poor carbon undergo cell cycle entry with a large difference in the ratio of Cln3 to Whi5 suggests that Whi5 protein concentration may not have a major influence on cell cycle entry. To investigate further, we carried out additional tests of how the concentration of Whi5 influences cell cycle entry and cell size. A previous study found that an additional copy of *WHI5* causes an increase in cell size (*Schmoller et al., 2015*). However, the effects of an additional copy of *WHI5* were only tested under nutrient-poor synthetic media conditions in cells that lack Bck2, a poorly understood inducer of cell cycle entry that becomes essential in cells that lack Cln3. In addition, the extra copy of *WHI5* was tagged, which could influence function, and it included both the promoter and part of the coding sequence of a gene neighboring *WHI5*.

To avoid potential effects of tags, we constructed an integrating vector that includes only the wild-type untagged *WHI5* gene with the normal upstream and downstream control regions. The *WHI5* construct fully rescued the size defects of *whi5Δ* (*Figure 3—figure supplement 1A*). Integration of the plasmid into wild-type cells to introduce an extra copy of *WHI5* had no effect on the size of cells growing in YP medium containing rich or poor carbon (*Figure 3A*). When cells were grown in nutrient-poor synthetic medium, the extra copy of *WHI5* caused a 7% increase in median cell size in rich carbon and a 10% increase in poor carbon (*Figure 3B*). We also tested the effects of a twofold decrease in dosage of *WHI5* by deleting one copy of *WHI5* in diploid cells and comparing the size of *WHI5/whi5Δ* cells to wild-type *WHI5/WHI5* cells in both rich and poor carbon media (*Figure 3C*). Loss of one copy of *WHI5* caused little change in cell size.

To test the effects of a larger increase in Whi5 protein levels, we put *WHI5* under the control of the *TEF1* promoter. Quantitative western blot analysis of Whi5-3XHA or Whi5-13MYC showed that the *TEF1* promoter drives a 12-fold increase in Whi5 protein levels (*Figure 3—figure supplement 1B*, and data not shown), as well as a gradual increase in Whi5 levels throughout G1 phase, which would prevent dilution (*Figure 3—figure supplement 1C and D*). Expression of untagged *WHI5* from the *TEF1* promoter increased median cell size by 35% in rich carbon medium and 58% in poor carbon medium (*Figure 3D*). A previous study found that overexpression of *WHI5* from the *GAL1* promoter in otherwise wild-type cells has only mild effects on cell size and cell cycle progression (*Costanzo et al., 2004*), while two other studies observed stronger effects of *GAL1-WHI5* (*de Bruin et al., 2004*; *Barber et al., 2020*).

It remains controversial whether Whi5 undergoes substantial dilution during G1 phase (*Dorsey et al., 2018*; *Litsios et al., 2019*). However, the maximal dilution that has been reported for most cells is 20-30% or less (*Schmoller et al., 2015*). Therefore, if dilution of Whi5 plays a role in cell size control one would expect a twofold increase or decrease in WHI5 dosage to have a substantial effect on cell size. The fact that twofold changes in *WHI5* dosage have little effect suggests that dilution of Whi5 is unlikely to play a major role in control of cell size. The finding that a twofold increase in *WHI5* has mild effects on cell size in nutrient-poor synthetic media is likely due to the large reduction in Cln3 protein levels in poor nutrients. The finding that a greater than 10-fold increase in Whi5 protein levels does not cause a lethal cell cycle arrest or a massive increase in cell size indicates that there must be dilution-independent mechanisms for overcoming Whi5 inhibition. Finally, the Whi5 dilution model cannot explain how cells in poor carbon initiate cell cycle entry despite a nearly 10-fold decrease in the ratio of Cln3 to Whi5. A recent study that utilized a heterologous promoter to drive Whi5 synthesis during G1 phase also reached the conclusion that dilution of Whi5 does not play a significant role in cell size control (*Barber et al., 2020*).

## Blocking membrane trafficking events required for cell growth prevents accumulation of Cln3

Together, the preceding experiments suggest that changes in the concentration of Whi5 do not play a substantial role in mechanisms that link cell cycle entry to cell growth. Rather, the data are more

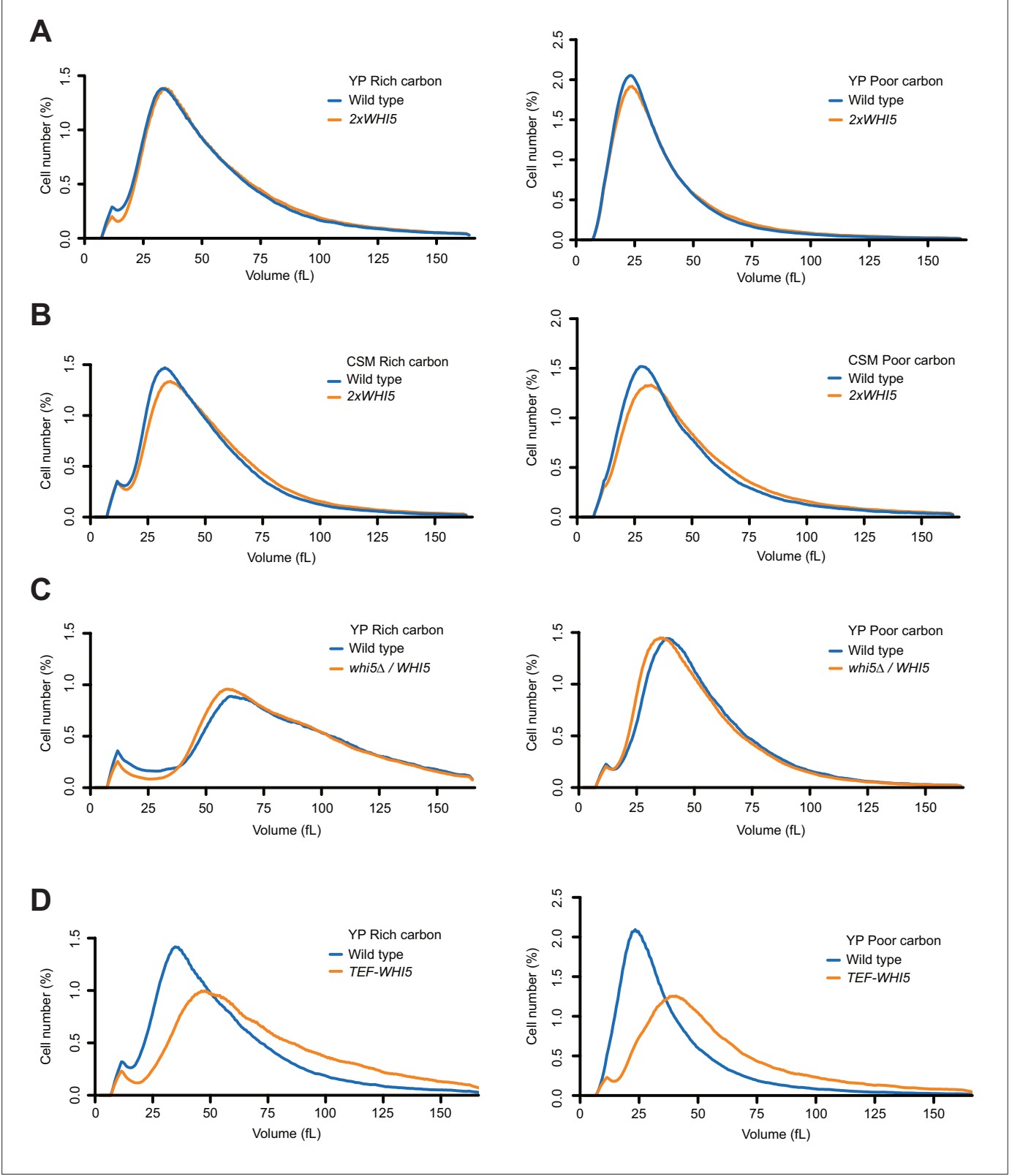

**Figure 3.** Increased dosage of Whi5 does not cause strong effects on cell size. (**A**) Wild-type and *2xWHI5* cells were grown to log phase in rich (YPD) or poor carbon (YPG/E), and cell size was measured using a Coulter counter. (**B**) Wild-type and *2xWHI5* cells were grown to log phase in complete synthetic medium containing dextrose (CSM-rich carbon) or glycerol/ethanol (CSM-poor carbon). Cell size was measured using a Coulter counter. (**C**) Wild-type diploid cells and heterozygous *WHI5/whi5Δ* cells were grown to log phase in rich (YPD) or poor carbon (YPG/E), and cell size was measured

*Figure 3 continued on next page*

*Figure 3 continued*

using a Coulter counter. (**D**) Wild-type and *TEF1-WHI5* cells were grown to log phase in rich (YPD) or poor carbon (YPG/E), and cell size was measured using a Coulter counter.

The online version of this article includes the following source data and figure supplement(s) for figure 3:

**Figure supplement 1.** Increased dosage of Whi5 does not cause strong effects on cell size.

**Figure supplement 1—source data 1.** Source data for *Figure 3—figure supplement 1B*.

**Figure supplement 1—source data 2.** Source data for *Figure 3—figure supplement 1B*.

**Figure supplement 1—source data 3.** Source data for *Figure 3—figure supplement 1C*. This panel also shows data for Cln3-6XHA, which is not discussed.

**Figure supplement 1—source data 4.** Source data for *Figure 3—figure supplement 1C*.

consistent with a model in which accumulation of Cln3 could be the critical readout of growth that triggers cell cycle entry. We therefore investigated the relationship between accumulation of Cln3 and cell growth. As a first step, we sought to test whether accumulation of Cln3 protein in G1 phase is dependent upon growth. To do this, we first searched for ways to enforce a rapid block to growth during G1 phase. In budding yeast, vesicular traffic that drives bud growth after cell cycle entry occurs along actin cables, and depolymerization of actin causes rapid cessation of bud growth. However, we discovered that addition of the actin depolymerizing drug latrunculin A had no effect on growth during G1 phase, even though bud emergence was completely blocked. We considered the possibility that growth in G1 phase requires vesicular transport on microtubules, but addition of both latrunculin A and the microtubule depolymerizing drug nocodazole had no effect on cell growth in G1 phase (*Figure 4A and B*).

We next tested whether proteins that drive early steps in membrane trafficking pathways are required for growth in G1 phase. Sec7 is required for ER-to-Golgi transport (*Lupashin et al., 1996*; *Deitz et al., 2000*). Inactivation of Sec7 using an auxin-inducible degron (sec7-AID) caused a rapid and complete block of growth in G1 phase (*Figure 4C*), as well as a complete block of bud emergence (*Figure 4D*). Inactivation of Sec7 completely blocked the gradual accumulation of Cln3 protein that normally occurs during growth in G1 phase (*Figure 4E*). We obtained similar results using a temperature-sensitive allele of SEC7 (*sec7-1*) (not shown).

One explanation for the effects of inactivating Sec7 could be that blocking membrane traffic causes a general decrease in the rate of protein synthesis, which would lead to a failure to accumulate Cln3. However, previous work found that protein synthesis continues after membrane traffic is blocked (*Novick and Schekman, 1979*). Moreover, we found that levels of a loading control protein increased normally, which argues against a general decrease in the rate of protein synthesis (*Figure 4E*, *Figure 4—figure supplement 1A*). Finally, we directly measured the rate of protein synthesis in *SEC7-AID* cells via incorporation of $^{35}$S-methionine. To do this, auxin was added to asynchronously growing *SEC7-AID* cells and the rate of incorporation of $^{35}$S-methionine was measured during 15 min intervals after addition of auxin (*Figure 4F*). We detected a few changes in the pattern of proteins synthesized in the *SEC7-AID* cells, which is likely caused by a failure in post-translational processing of proteins in the secretory pathway. However, we did not detect a decrease in the rate of protein synthesis. We obtained similar results when we induced destruction of *SEC7-AID* in synchronized cells undergoing growth in G1 phase (*Figure 4—figure supplement 1B–D*). These data suggest that failure to accumulate Cln3 after inactivation of Sec7 is not due to a general decrease in the rate of protein synthesis.

Together, these observations show that the gradual rise in Cln3 levels during G1 phase is dependent upon membrane trafficking events that drive plasma membrane growth, consistent with the idea that Cln3 levels could provide a readout of cell growth.

## The budding yeast homologs of mammalian SGK kinases are required for accumulation of Cln3 during G1 phase

To further explore the relationship between cell growth and gradual accumulation of Cln3, we searched for signals that influence cell growth and size in G1 phase, with the goal of testing whether these signals also influence Cln3 accumulation. Polar bud growth that occurs after G1 phase requires CDK activity (*McCusker et al., 2007*). We therefore first tested whether CDK activity is required for

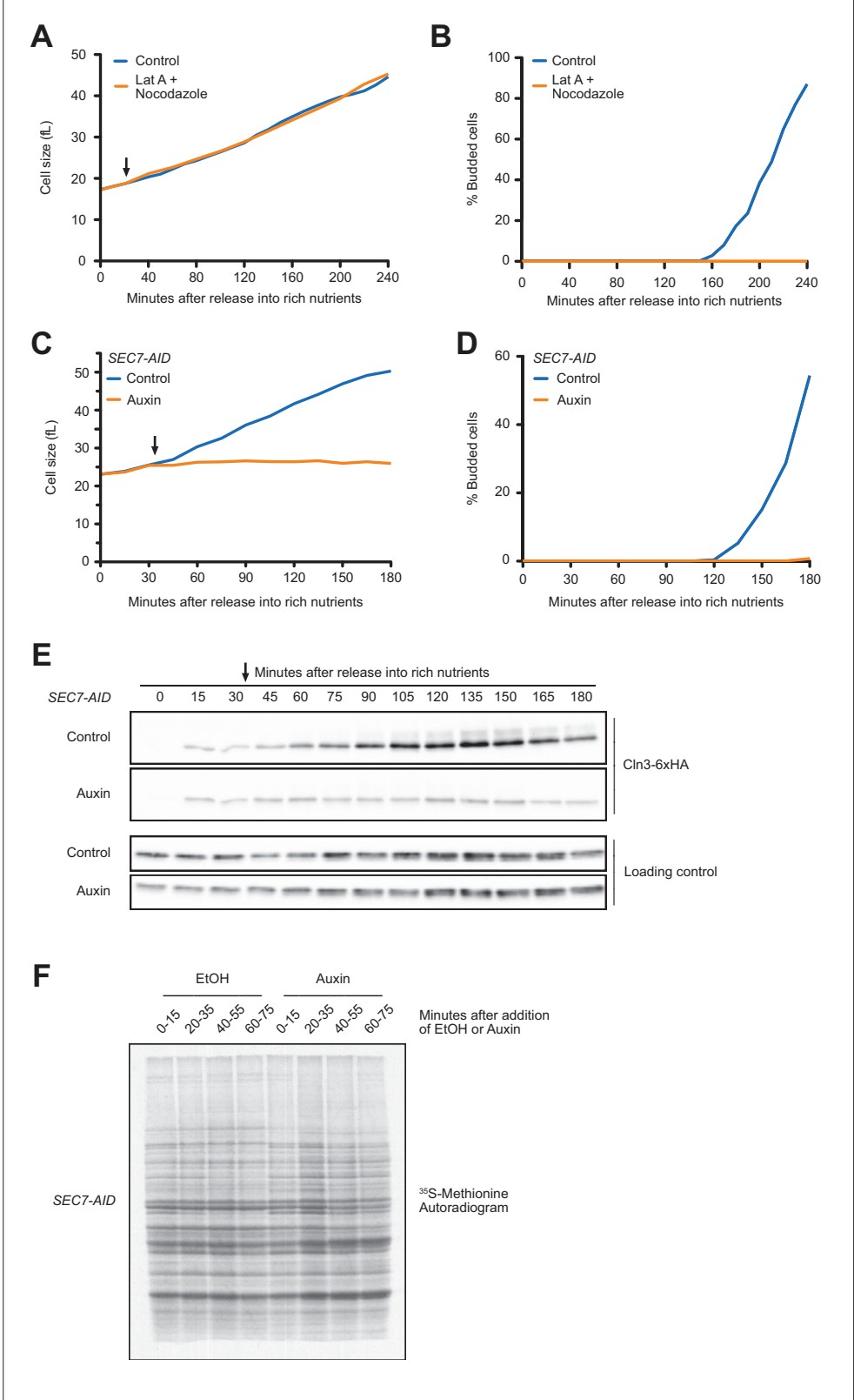

**Figure 4.** Blocking membrane trafficking events required for cell growth prevents accumulation of Cln3. (**A, B**) Wild-type cells were grown to mid-log phase in poor carbon (YPG/E), and small unbudded cells were isolated by centrifugal elutriation. The cells were split into two cultures and were then released into rich carbon (YPD) at 25°C. After 20 min, 100 µM of latrunculin A and 20 µM nocodazole were added to one culture (arrow). (**A**) Median cell

*Figure 4 continued on next page*

*Figure 4 continued*

size was measured at 20 min intervals using a Coulter counter and plotted as a function of time. (**B**) The percentage of budded cells as a function of time. (**C–E**) *SEC7-AID* cells were grown to mid-log phase in poor carbon, and small unbudded cells were isolated by centrifugal elutriation. The cells were split and then released into rich carbon at 25°C. After 30 min, 1 mM auxin was added to one culture (arrow). (**C**) Median cell size was measured at 15 min intervals using a Coulter counter and plotted as a function of time. (**D**) The percentage of budded cells as a function of time. (**E**) The behavior of Cln3-6XHA in the *SEC7-AID* cells was analyzed by western blot. An anti-Nap1 antibody was used as a loading control. (**F**) Autoradiogram of $^{35}$S-methionine labeling to detect de novo protein synthesis. *SEC7-AID* cells were grown to mid-log phase in -MET synthetic media containing dextrose. After auxin or vehicle addition, samples were labeled with $^{35}$S-methionine for 15 min intervals starting every 20 min to measure the rate of protein synthesis within the 15 min intervals.

The online version of this article includes the following figure supplement(s) for figure 4:

**Source data 1.** Source data for *Figure 4E*.

**Source data 2.** Source data for *Figure 4E*.

**Source data 3.** Source data for *Figure 4F*.

**Figure supplement 1.** Blocking membrane trafficking events required for cell growth prevents accumulation of Cln3.

**Figure supplement 1—source data 1.** Source data for *Figure 4—figure supplement 1D*.

growth in G1 phase. There are two CDKs that play overlapping roles in G1 phase: Cdk1 and Pho85. We utilized a strain that is dependent upon analog-sensitive alleles of both kinases (*cdk1-as pho85-as*) to test whether CDKs are required for growth in G1 phase. Inhibition of both kinases had no effect on growth in G1 phase (***Figure 5—figure supplement 1A***). This observation, along with the discovery that actin is not required for growth in G1 phase (***Figure 4A***), suggests that there may be substantial differences between the mechanisms that drive growth in G1 phase and those that drive growth of the daughter bud.

We next tested Tor kinase signaling pathways, which play conserved roles in control of cell growth. Tor kinases are assembled into two distinct multi-protein signaling complexes called TORC1 and TORC2 (***Loewith et al., 2002***). A key downstream target of TORC1 is the Sch9 kinase, a member of the AGC kinase family that is thought to be the functional ortholog of vertebrate S6 kinase. Sch9 mediates TORC1-dependent control of ribosome biogenesis (***Urban et al., 2007***). Cells that lack Sch9 are viable but proliferate slowly and show a large decrease in cell size (***Jorgensen et al., 2002***; ***Jorgensen and Tyers, 2004a***). To investigate the effects of a loss of function of Sch9, we utilized an analog-sensitive allele of *SCH9* (*sch9-as*) (***Jorgensen and Tyers, 2004a***). Previous work found that *sch9-as* cells are smaller than wild-type cells in the absence of inhibitor, consistent with the fact that most analog-sensitive kinases have partially compromised function. Addition of the inhibitor makes the cells very small, consistent with complete or nearly complete inhibition of sch9-as kinase activity (***Jorgensen and Tyers, 2004aJorgensen and Tyers, 2004a***). Inhibition of *sch9-as* in G1 phase had no effect on growth rate (***Figure 5A***) but caused a slight delay in bud emergence (***Figure 5B***). Inhibition of sch9-as also caused a reduction in Cln3 levels but did not block gradual accumulation of Cln3 during growth in G1 phase (***Figure 5C***). Addition of inhibitor to wild-type cells did not have substantial effects on wild-type control cells (***Figure 5—figure supplement 1B and C***). The finding that acute inhibition of sch9-as does not have strong immediate effects on growth rate and cell size suggests that the effects of *sch9Δ* could be a long-term consequence of a decreased rate of ribosome biogenesis. A caveat is that the sch9-as protein could retain a low level of activity in the presence of inhibitor.

We next tested whether components of a TORC2 signaling network are required for growth in G1 phase. A key function of TORC2 is to activate a pair of redundant kinase paralogs called Ypk1 and Ypk2, which are the budding yeast homologs of vertebrate SGK kinases (***Casamayor et al., 1999***; ***Kamada et al., 2005***). Previous work found that Ypk1/2 are required for normal control of cell growth and size (***Lucena et al., 2018***). For example, loss of Ypk1 causes a large decrease in cell size, as well as a reduced rate of proliferation. Loss of both Ypk1 and Ypk2 is lethal.

To test whether Ypk1/2 influence cell growth and accumulation of Cln3 in G1 phase, we utilized an analog-sensitive allele of *YPK1* in a *ypk2Δ* background (*ypk1-as ypk2Δ*) (***Sun et al., 2012***). Inhibition of ypk1-as in early G1 phase caused a decrease in growth rate and a complete failure in bud emergence (***Figure 6A and B***). It also caused a rapid and complete loss of Cln3 protein (***Figure 6C***). In wild-type

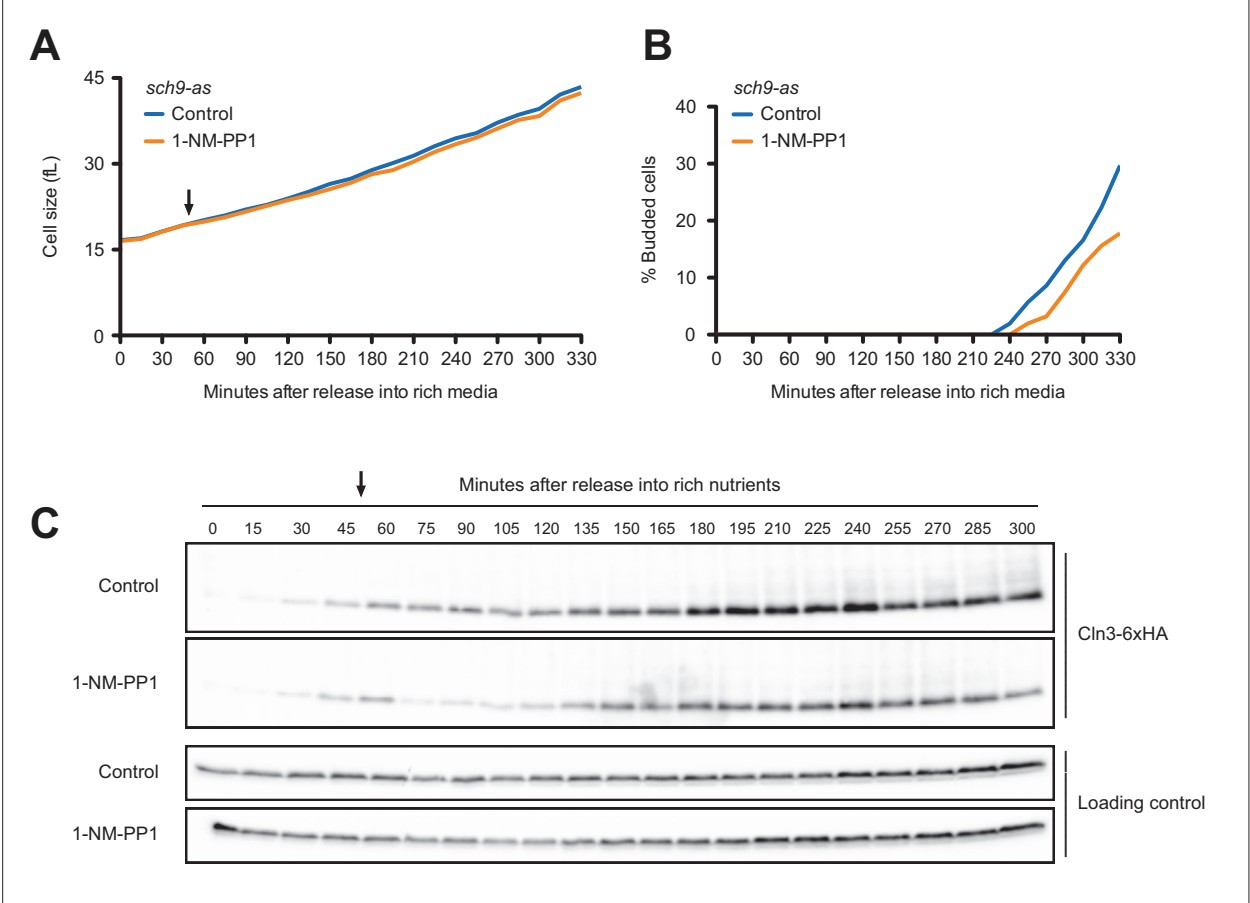

**Figure 5.** Inhibition of Sch9 does not cause strong effects on accumulation of Cln3 during G1 phase.
 *sch9-as* cells were grown to mid-log phase in poor carbon (YPG/E), and small unbudded cells were isolated via centrifugal elutriation. The cells were split into two cultures and were released into rich carbon (YPD, not supplemented with additional adenine; see Materials and methods). After 45 min, 250 nM 1-NM-PP1 was added to one culture (arrow). (**A**) Median cell size was measured at 15 min intervals using a Coulter counter and plotted as a function of time. (**B**) The percentage of budded cells was plotted as a function of time. (**C**) The behavior of Cln3-6XHA was analyzed by western blot. An anti-Nap1 antibody was used for a loading control.

The online version of this article includes the following figure supplement(s) for figure 5:

**Source data 1.** Source data for *Figure 5C*.

**Source data 2.** Source data for *Figure 5C*.

**Figure supplement 1.** Inhibition of cyclin-dependent kinases does not cause strong effects on growth or accumulation of Cln3 during G1 phase.

**Figure supplement 1—source data 1.** Source data for *Figure 5—figure supplement 1C*.

control cells, the inhibitor caused a slight decrease in growth rate and a slight decrease in Cln3 levels (*Figure 6—figure supplement 1A–C*). The loss of Cln3 after inhibition of ypk1-as did not appear to be caused by a general shutdown of protein synthesis because growth continued and levels of a loading control protein increased (*Figure 6A and B*, *Figure 6—figure supplement 1D*). Direct measurement of the rate of protein synthesis confirmed that inhibition of ypk1-as did not cause a general shutdown of protein synthesis (*Figure 6—figure supplement 1E*). Previous work found that *ypk1Δ* alone causes a large decrease in cell size (*Lucena et al., 2018*). Here, we found that *ypk1Δ* caused a large reduction in Cln3 levels in asynchronous cells in both rich and poor carbon (*Figure 6D*).

Ypk1/2 undergo complex regulation and are phosphorylated by multiple kinases. We found that changes in Cln3 protein levels that occur shortly after a shift from rich to poor carbon closely paralleled changes in the phosphorylation state of Ypk1 that can be detected via shifts in electrophoretic mobility (*Figure 6E*). Thus, the transient increase in Cln3 levels immediately after a shift to poor carbon was correlated with a loss of Ypk1 phosphorylation, and the decrease in Cln3 levels starting at

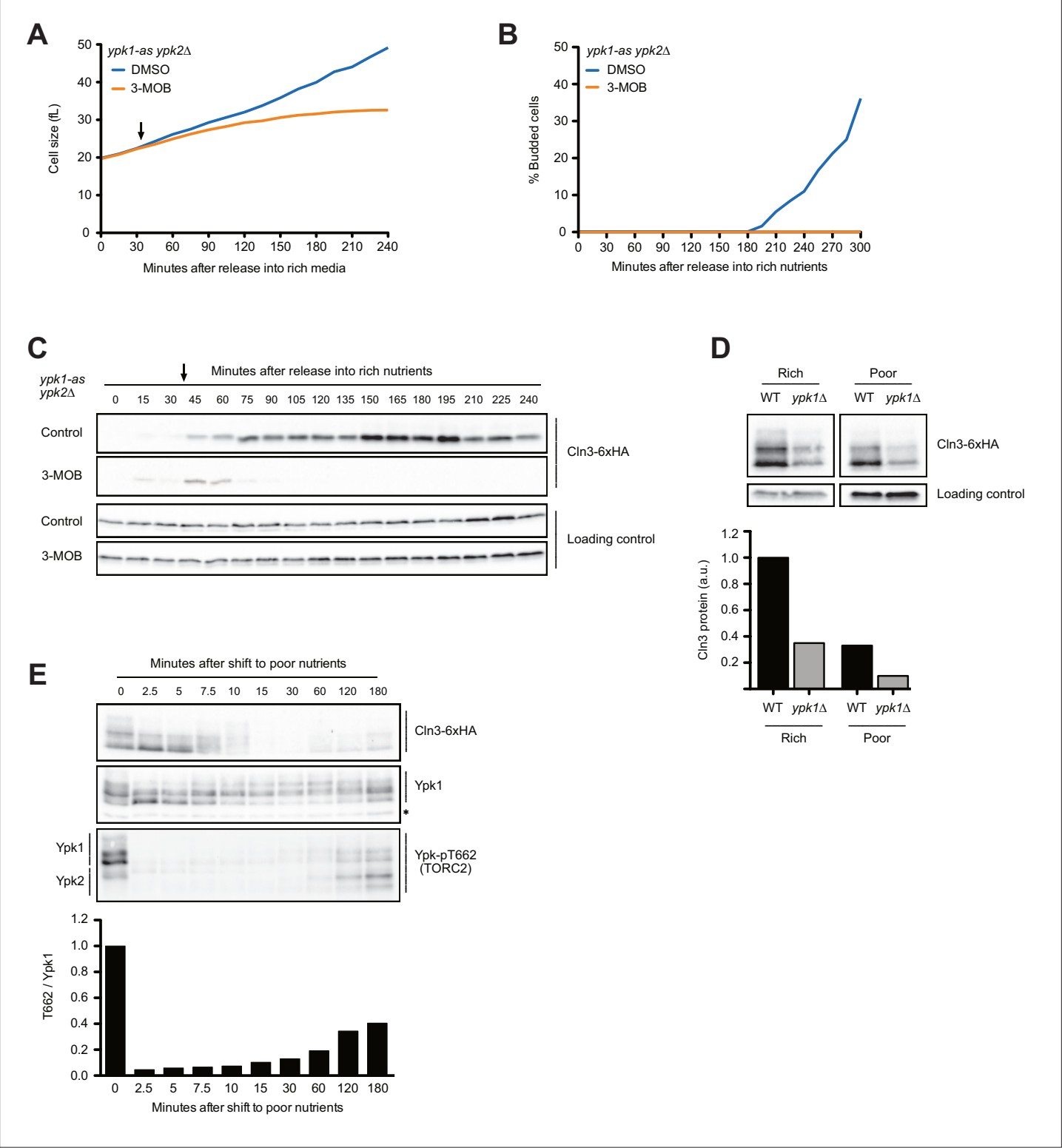

**Figure 6.** The budding yeast homologs of mammalian SGK kinases are required for accumulation of Cln3 during G1 phase. (**A–C**) *ypk1-as ypk2Δ* cells were grown overnight in poor carbon (YPG/E) to mid-log phase, and small unbudded cells were isolated by centrifugal elutriation. The cells were divided into two cultures and were then released into rich carbon (YPD, not supplemented with additional adenine; see Materials and methods). After 30 min, 25 µM 3-MOB-PP1 was added to one culture (arrow). (**A**) Median cell size was measured at 15 min intervals using a Coulter counter and plotted as a function of time. (**B**) The percentage of budded cells as a function of time. (**C**) The behavior of Cln3-6XHA in the *ypk1-as ypk2Δ* cells was analyzed by western blot. An anti-Nap1 antibody was used for a loading control. (**D**) Wild-type and *ypk1Δ* cells were grown to mid-log phase in rich or poor

*Figure 6 continued on next page*

*Figure 6 continued*

carbon. The levels of Cln3-6XHA protein were analyzed by western blot. An anti-Nap1 was used for a loading control. Levels of Cln3-6XHA protein were quantified relative to levels of Cln3-6XHA in wild-type cells in rich carbon. Cln3 protein levels were first normalized to the loading control. (**E**) Wild-type cells were grown to mid-log phase in rich carbon (YPD) and then shifted to poor carbon (YPG/E) at 25°C. The behavior of Cln3-6XHA, TORC2-dependent phosphorylation of Ypk1/2, and total Ypk1 protein was assayed by western blot. A phospho-specific antibody was used to detect a TORC2-dependent phosphorylation site present on both Ypk1 and Ypk2 (referred to as anti-Ypk-pT662). Total Ypk1 protein was detected with an anti-Ypk1 antibody. Asterisk indicates a background band that also serves as a loading control.

The online version of this article includes the following figure supplement(s) for figure 6:

**Source data 1.** Source data for *Figure 6C*.

**Source data 2.** Source data for *Figure 6C*.

**Source data 3.** Source data for *Figure 6D*.

**Source data 4.** Source data for *Figure 6D*.

**Source data 5.** Source data for *Figure 6D*.

**Source data 6.** Source data for *Figure 6D*.

**Source data 7.** Source data for *Figure 6E*.

**Source data 8.** Source data for *Figure 6E*.

**Source data 9.** Source data for *Figure 6E*.

**Figure supplement 1.** The budding yeast homologs of mammalian SGK kinases are required for accumulation of Cln3 during G1 phase.

**Figure supplement 1—source data 1.** Source data for *Figure 6—figure supplement 1B*.

**Figure supplement 1—source data 2.** Source data for *Figure 6—figure supplement 1B*.

**Figure supplement 1—source data 3.** Source data for *Figure 6—figure supplement 1E*.

10 min was correlated with an increase in Ypk1 phosphorylation. A reappearance of Cln3 at the end of the time course was accompanied by another decrease in Ypk1 phosphorylation. The phosphorylation of Ypk1 that can be detected by electrophoretic mobility shifts appears to be due primarily to related kinase paralogs called Fpk1 and Fpk2; however, the functions of these phosphorylation events are poorly understood (*Roelants et al., 2010*). Nevertheless, the correlation between changes in Ypk1 phosphorylation and changes in Cln3 protein levels provides further correlative evidence for a connection between Ypk1 signaling and Cln3 protein levels.

We also assayed TORC2-dependent phosphorylation of Ypk1 and Ypk2 in the same samples, which can be detected with a phospho-specific antibody that recognizes a site found on both kinases (referred to as T662 in Ypk1) (*Niles et al., 2012*). It is unknown whether TORC2-dependent phosphorylation influences the electrophoretic mobility of Ypk1/2, and TORC2-dependent phosphorylation of Ypk1/2 is measured as the intensity of the signal generated by the anti-pT662 antibody. Previous studies found that a shift from rich to poor carbon causes rapid loss of TORC2-dependent phosphorylation of Ypk1/2 (*Lucena et al., 2018*). Here, we found that loss of TORC2-dependent phosphorylation of Ypk1/2 upon a shift to poor carbon was immediate and preceded the decrease in Cln3 levels by approximately 10 min (*Figure 6E*, bottom panel). The increase in Cln3 phosphorylation that occurred later in the time course as cells adapted to the new carbon source was correlated with an increase in TORC2-dependent phosphorylation of Ypk1/2. Thus, changes in Cln3 levels were not strongly correlated with TORC2-dependent phosphorylation of Ypk1/2.

Together, the data show that Ypk1/2, which were previously shown to play roles in control of cell growth and size, also play a role in regulating levels of Cln3. The data further suggest that Cln3 protein levels are unlikely to be influenced only by translation rate. Rather, the data indicate that poorly understood signals associated with plasma membrane growth and the TORC2 network strongly influence Cln3 levels.

## Sphingolipid-dependent signals influence Cln3 levels

A key function of Ypk1/2 is to control the activity of a biosynthetic pathway that builds sphingolipids and ceramide lipids (*Aronova et al., 2008*; *Roelants et al., 2011*; *Sun et al., 2012*; *Muir et al., 2014*). A simplified overview of the role of Ypk1/2 is shown in *Figure 7—figure supplement 1A*. Ypk1/2 stimulate the enzyme that initiates production of sphingolipids and also stimulates ceramide synthase, which builds ceramide from sphingolipid precursors. Sphingolipids and ceramides play

poorly understood roles in signaling. In previous work, we found that Ypk1/2-dependent control of the ceramide synthesis pathway is required for normal control of cell growth and size (*Lucena et al., 2018*). For example, myriocin, a small molecule inhibitor of sphingolipid synthesis, causes a dose-dependent decrease in growth rate in G1 phase and a corresponding decrease in cell size at cell cycle entry. Similarly, loss of ceramide synthase causes a large reduction in cell size, as well as a complete failure in nutrient modulation of cell size, growth rate, and TORC2 activity.

To test whether Ypk1/2 control Cln3 levels via sphingolipids or ceramides, we first tested the effects of myriocin, which inhibits the first step in production of sphingolipids. In previous work, we analyzed the effects of sublethal doses of myriocin on cell growth and size in G1 phase (*Lucena et al., 2018*). Here, we used a higher concentration of myriocin that blocks proliferation. Addition of myriocin caused a large reduction in growth rate in G1 phase and delayed bud emergence but caused only a modest decrease in Cln3 levels (*Figure 7A–D*). Myriocin also caused cells to initiate bud emergence at a smaller size (*Figure 7C*). Thus, the complete loss of Cln3 caused by inhibition of Ypk1/2 is not due to a loss of sphingolipid-dependent signals.

In previous work, we found that ceramide-dependent signals modulate TORC2 activity via a feedback loop and are also required for normal control of cell growth and size (*Lucena et al., 2018*; *Alcaide-Gavilán et al., 2018*). We therefore also tested the effects of inactivating ceramide synthase, which builds ceramide from sphingolipid precursors. The catalytic subunit of ceramide synthase is encoded by a pair of redundant paralogs called *LAC1* and *LAG1* (*Figure 7—figure supplement 1A*). In contrast to myriocin, we found that *lac1Δ lag1Δ* caused a substantial decrease in levels of Cln3 in asynchronous cells (*Figure 7E*). This was surprising because both myriocin and inactivation of ceramide synthase block production of ceramide, yet they appear to have different effects on Cln3. An explanation could be that inactivation of ceramide synthase leads to a build-up of sphingolipids that inhibit production of Cln3. To test this, we added exogenous sphingolipids to cells. In previous work, we found that the sphingolipid phytosphingosine (PHS) causes a rapid and dramatic decrease in TORC2 signaling to Ypk1/2 (*Lucena et al., 2018*). The decrease in TORC2 signaling was dependent upon ceramide synthase, which provided evidence for ceramide-dependent feedback signaling to TORC2. Here, we found that exogenously added PHS caused a rapid loss of Cln3 in both wild-type and *lac1Δ lag1Δ* cells (*Figure 7F*). Levels of Cln3 recovered in wild-type cells, most likely due to conversion of the added PHS to ceramide. In contrast, Cln3 levels did not recover in *lac1Δ lag1Δ* cells.

These data show that sphingolipid-dependent signals strongly influence Cln3 levels and provide further evidence that complex signaling mechanisms control accumulation of Cln3 protein.

## Discussion

### The concentration of Cln3 protein increases during growth in G1 phase

One class of models for cell size control suggests that accumulation of Cln3 is a key reporter of cell growth during G1 phase (*Jorgensen and Tyers, 2004b*; *Turner et al., 2012*). However, there have been conflicting reports about the behavior of Cln3 in G1 phase (*Zapata et al., 2014*; *Schmoller et al., 2015*; *Litsios et al., 2019*). Here, we used quantitative western blotting to analyze the behavior of Cln3 during growth in G1 phase in rich versus poor carbon. Under the growth conditions used for our experiments, we found that the number of Cln3 molecules per cell increases approximately 10-fold before cell cycle entry in rich carbon and 2.5-fold in poor carbon. When combined with growth data, these data suggest that Cln3 concentration increases sevenfold before bud emergence in rich carbon and twofold in poor carbon. The large increase in Cln3 concentration observed for cells growing in rich carbon may be due partially to the fact that the cells were grown in poor carbon prior to synchronization and therefore underwent an unusually extensive growth interval during G1 phase to reach the larger size that is characteristic of cells in rich carbon.

In both rich and poor carbon, synthesis of Cln3 protein outpaces the overall rate of protein synthesis that would be needed for the 20–30% increase in cell size that occurs during growth in the same interval. Thus, increasing Cln3 concentration could generate a signal with substantial dynamic range that could be used to measure growth and trigger cell cycle entry. Similar conclusions were reached in a previous study that used an in vivo reporter to measure Cln3 translation rates and targeted mass spectrometry to measure Cln3 protein levels (*Litsios et al., 2019*).

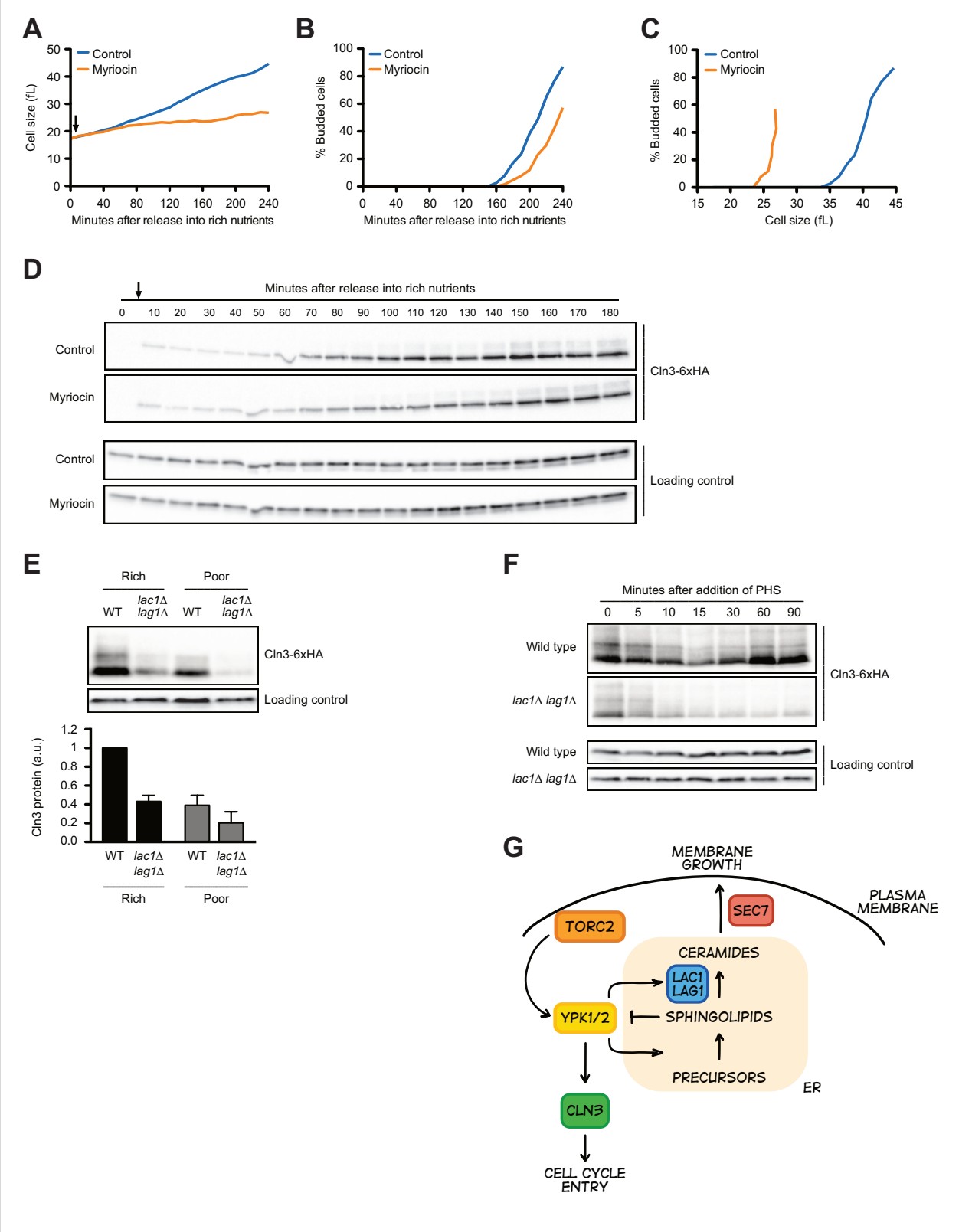

**Figure 7.** Sphingolipid-dependent signals influence Cln3 levels. (**A–D**) Wild-type cells were grown overnight in poor carbon (YPG/E), and small unbudded cells were isolated via centrifugal elutriation. The cells were split into two cultures and released into rich carbon (YPD) at 25°C. After taking the initial time point, 5 µg/mL myriocin was added to one culture (arrow). (**A**) Median cell size was measured at 20 min intervals using a Coulter counter and plotted as a function of time. (**B**) The percentage of budded cells as a function of time. (**C**) The percentage of budded cells plotted as a function of

*Figure 7 continued on next page*

Figure 7 continued

cell size at each time point. (**D**) The levels of Cln3-6XHA protein were analyzed by western blot. An anti-Nap1 antibody was used for a loading control. (**E**) Wild-type and *lac1Δ lag1Δ* cells were grown to mid-log phase in rich (YPD) and poor carbon (YPG/E). The levels of Cln3-6xHA protein were analyzed by western blot. An anti-Nap1 antibody was used as a loading control. Levels of Cln3-6xHA protein were quantified relative to levels of Cln3-6xHA in wild-type cells in rich carbon. Cln3 protein levels were first normalized to the loading control. Error bars represent SEM of three biological replicates. (**F**) Wild-type and *lac1Δ lag1Δ* cells were grown to mid-log phase in rich carbon (YPD). 20 μM phytosphingosine (PHS) was added to each culture, and the cultures were incubated at 25°C. The levels of Cln3-6xHA protein were analyzed by western blot. An anti-Nap1 antibody was used as a loading control. (**G**) A model depicting how Cln3 levels could be modulated by Ypk1/2.

The online version of this article includes the following figure supplement(s) for figure 7:

**Source data 1.** Source data for *Figure 7D*.

**Source data 2.** Source data for *Figure 7D*.

**Source data 3.** Source data for *Figure 7E*.

**Source data 4.** Source data for *Figure 7E*.

**Source data 5.** Source data for *Figure 7F*.

**Source data 6.** Source data for *Figure 7F*.

**Figure supplement 1.** A simplified model of the mechanisms by which Ypk1/2 control synthesis of sphingolipids and ceramide.

## The threshold amount of Cln3 required for cell cycle entry is dramatically reduced in poor carbon

Simple models based on previous studies would suggest that cell cycle entry at a reduced cell size in poor carbon could be achieved by reducing levels of Whi5 protein or by increasing levels of Cln3. However, we found that Cln3 levels are reduced nearly 10-fold in poor carbon, while levels of Whi5 do not change. Thus, cells in poor carbon undergo cell cycle entry despite a nearly 10-fold decrease in the ratio of Cln3 to Whi5. We also found that Cln3 protein levels are exquisitely sensitive to changes in carbon source that influence growth rate. Thus, a shift from rich to poor carbon, which causes a rapid decrease in growth rate, causes a transient increase in Cln3 protein levels within 5 min, followed by a dramatic reduction in Cln3 levels. These data show that Cln3 is highly responsive to nutrient-dependent signals that control cell growth and size, as expected for a critical regulator of cell size in G1 phase.

One estimate suggests that there could be as few as 100 molecules of Cln3 per cell in rich carbon (*Cross et al., 2002*), which would suggest that there could be as few as 10–20 molecules of Cln3 per cell in poor carbon. In contrast, Whi5 could be present at approximately 2500 molecules per cell, although there is considerable variance in estimates of the abundance of Whi5 protein (*Ho et al., 2018*; *Dorsey et al., 2018*). The mechanism that sets the threshold amount of Cln3 required to overcome Whi5 inhibition is key to understanding how cell cycle progression is linked to cell growth yet remains one of the central enigmas of cell size control. Recent work suggests that the transcription factors that promote cell cycle entry are present at higher concentrations in poor carbon, which could help promote cell cycle entry at lower levels of Cln3 (*Dorsey et al., 2018*).

## The concentration of Whi5 protein has little influence on cell size

The effects of Whi5 overexpression on cell size have been the primary mechanistic test of the Whi5 dilution model; however, increased dosage of Whi5 could also influence a positive feedback loop in which the late G1 cyclins Cln1 and Cln2 promote cell cycle entry via inactivation of Whi5 (*Cross and Tinkelenberg, 1991*). Thus, current data do not distinguish whether any effects of Whi5 overexpression are due to inhibition of the initial initiation of late G1 cyclin transcription versus inhibition Whi5 inactivation via positive feedback.

To further test the Whi5 dilution model, we investigated the effects of increased or decreased expression of *WHI5* on cell size. An extra copy of untagged Whi5 expressed from its own promoter had little effect on cell size in wild-type cells growing in YP media containing rich or poor carbon. Similarly, loss of one copy of *WHI5* in diploid cells had little effect on cell size. We further found that greater than 10-fold overexpression of *WHI5* from the *TEF1* promoter had a relatively modest effect on cell size in rich carbon, which indicates that there are dilution-independent mechanisms for inactivation of Whi5. Furthermore, a key test of the model is to determine the effects of synthesizing Whi5

during G1 phase, which would prevent dilution. A previous study tested this by using a heterologous promoter to drive Whi5 synthesis during G1 phase and did not observe the effects predicted by the Whi5 dilution model (**Barber et al., 2020**). We conclude that growth-dependent dilution of Whi5 is unlikely to play a major role in measuring cell size in G1 phase. Multiple recent studies reached similar conclusions (**Dorsey et al., 2018**; **Litsios et al., 2019**; **Barber et al., 2020**).

Dilution models for cell size control present a number of unresolved mechanistic issues. For example, dilution models require that key factors show a concentration-dependent activity. The only known activity of Whi5 is to bind and inhibit the SBF transcription factor. For Whi5 to repress transcription at SBF promoters, it must be present in excess over SBF binding sites and bind with sufficient affinity to achieve high occupancy at promoters. Moreover, to achieve high occupancy at SBF promoters, Whi5 must be present at a concentration higher than the $K_d$ for its interaction with SBF. One study estimated the concentration of Whi5 in the nucleus to be 120 nM (**Dorsey et al., 2018**). Independent studies that measured the total number of Whi5 molecules in the cell and nuclear volume suggest that the concentration could be much higher (**Jorgensen et al., 2007**; **Ho et al., 2018**). The $K_d$ for the interaction between Whi5 and SBF is unknown but could easily be in the low nM range. For comparison, the Rb transcription inhibitor binds to the E2F transcription factor with a $K_d$ of 40 nM (**Burke et al., 2010**). Thus, the concentration of Whi5 in the nucleus may be substantially higher than the $K_d$ for binding to SBF. In this case, a 20–30% change in cell volume would have no effect on occupancy of Whi5 at SBF promoters, which is the only known activity of Whi5. It has been proposed that growth-dependent dilution causes a change in the ratio of Cln3 to Whi5. However, under the poor carbon conditions used to test the dilution model, the concentration of Whi5 is clearly much higher than the concentration of Cln3, so a less than twofold change in Whi5 concentration will have little effect on the ratio. Phosphorylation of Whi5 by Cdk1 could reduce the affinity of Whi5 for SBF, but in that case there may be no reason to hypothesize that anything other than rising Cln3 levels leads to inactivation of Whi5. It is certainly possible that there are biochemical parameters for the mechanisms that inactivate Whi5 that would be compatible with dilution models. However, in the absence of any data on those parameters it remains unclear whether dilution models are mechanistically viable.

## Accumulation of Cln3 during G1 phase is dependent upon membrane trafficking events that drive cell growth

Previous studies have proposed that accumulation of Cln3 protein could be a key readout of cell growth. To provide a measure of growth, accumulation of Cln3 must be dependent upon growth. To test this, we used an auxin-inducible degron allele of *SEC7* to block membrane trafficking events that drive cell growth. Inactivation of Sec7 in early G1 phase blocked cell growth, as well as accumulation of Cln3. This observation, combined with the results of our analysis of Cln3 accumulation in wild-type cells, suggests that gradual accumulation of Cln3 in G1 phase is dependent upon growth and correlated with the extent of growth, as expected for a protein that provides a readout of growth.

In previous work, we found additional evidence that membrane trafficking events that drive cell growth generate signals that are correlated with growth, which could be used to measure growth. For example, delivery of vesicles to the plasma membrane in growing daughter buds generates signals that lead to phosphorylation of the kinase Pkc1 as well as two related kinases called Gin4 and Hsl1 (**Anastasia et al., 2012**; **Jasani et al., 2020**). In each case, multi-site phosphorylation of these kinases appears to gradually increase during bud growth, and phosphorylation is dependent upon growth and correlated with the extent of growth. The signals relayed by these kinases are required for normal regulation of the duration and extent of bud growth, but there is no evidence that they influence growth in G1 phase. Several previous studies also found connections between membrane traffic, ribosome biogenesis, and cell size (**Mizuta and Warner, 1994**; **Nanduri and Tartakoff, 2001**; **Lempiäinen et al., 2009**; **Singh and Tyers, 2009**).

## Cln3 protein levels are controlled by the budding yeast homologs of mammalian SGK kinases

We also searched for signals that drive growth-dependent accumulation of Cln3. Growth of the daughter bud after cell cycle entry is dependent upon Cdk1 activity (**McCusker et al., 2007**). However, growth in G1 phase and accumulation of Cln3 showed no dependence on Cdk1 activity. Surprisingly, growth in G1 phase was also not dependent upon actin filaments, which are required for bud growth.

The fact that bud growth is strongly dependent upon Cdk1 and actin filaments, whereas growth in G1 phase is not, suggests that there may be substantial differences in the mechanisms that drive cell growth at different stages of the cell cycle.

Inhibition of Ypk1/2, the budding yeast homologs of human SGK kinases, caused a rapid and complete loss of Cln3 protein. Previous studies have shown that Ypk1/2 relay nutrient-dependent signals and are required for normal control of cell growth and size (*Lucena et al., 2018*). Thus, the discovery that Ypk1/2 control Cln3 protein levels establishes a link between Ypk1/2 and Cln3 that could help explain how Ypk1/2 and nutrient-dependent signals modulate cell size. The mechanism by which Ypk1/2 influence Cln3 protein levels is unknown. Since Cln3 is a rapid turnover protein, the mechanism could work at the level of transcription, translation, or protein stability. Further investigation of the mechanisms that link Cln3 levels to Ypk1/2 activity should provide important clues to how cell growth and size are controlled.

Ypk1/2 are direct targets of TORC2 and the TORC2 signaling network is strongly modulated by carbon source, which suggests that TORC2 could play a role in modulating Cln3 levels (*Kamada et al., 2005*; *Niles et al., 2012*; *Lucena et al., 2018*; *Alcaide-Gavilán et al., 2018*). However, there are additional kinases that phosphorylate Ypk1/2 and regulation of Ypk1/2 remains poorly understood (*Casamayor et al., 1999*; *Roelants et al., 2010*). Thus, it is also possible that there are signals that act in parallel with TORC2 to modulate Cln3 levels via Ypk1/2. A further complication is that Ypk1/2 relay poorly understood feedback signals that regulate their upstream kinases, which can make it difficult to clearly delineate direct versus indirect effects of manipulating the signaling network (*Roelants et al., 2010*; *Roelants et al., 2011*; *Berchtold et al., 2012*; *Lucena et al., 2018*; *Alcaide-Gavilán et al., 2018*).

There was an important difference between the effects of inhibiting Ypk1/2 and the effects of inhibiting Sec7. Inhibition of Ypk1/2 caused a reduced rate of growth and a rapid and complete loss of Cln3. In contrast, inhibition of Sec7 completely blocked cell growth as well as further accumulation of Cln3 but did not cause a loss of Cln3. These observations suggest that the effects of inhibiting Ypk1/2 are not a simple consequence of a slowdown in overall growth, and that Ypk1/2 could play a more direct role in controlling Cln3 levels. A key function of Ypk1/2 is to stimulate production of sphingolipids and ceramides, which relay signals that influence cell growth and size. Myriocin, an inhibitor of the first step of sphingolipid synthesis, did not block accumulation of Cln3. Similarly, loss of ceramide synthase reduced but did not eliminate Cln3. Thus, the disappearance of Cln3 upon inhibition of Ypk1/2 is not due to a failure in synthesis of sphingolipids and ceramides. Rather, another target of Ypk1/2 likely exists that influences Cln3 levels.

The different effects of Sec7, Ypk1/2 and myriocin on Cln3 levels are puzzling. Previous studies found that myriocin, inhibition of Ypk1/2, and inhibition of membrane traffic all cause increased TORC2-dependent phosphorylation of Ypk1/2, which is thought to lead to increased activity of Ypk1/2 (*Niles et al., 2014*; *Clarke et al., 2017*; *Lucena et al., 2018*). In each case, the increased TORC2-dependent phosphorylation of Ypk1/2 is thought to be due to poorly understood negative feedback loops in the TORC2 network. We confirmed that each of these experimental manipulations also cause increased TORC2-dependent phosphorylation of Ypk1/2 during growth in G1 phase (not shown). Although each experimental manipulation causes a decrease in growth rate, they each have different effects on accumulation of Cln3. We also found that addition of exogenous sphingolipids to cells causes a rapid decline in Cln3 levels that is not dependent upon conversion of sphingolipids to ceramides.

What kind of model could explain these observations? One possibility is that there is a pool of Ypk1/2 at the endoplasmic reticulum that promotes accumulation of Cln3, and sphingolipids produced at the endoplasmic reticulum inhibit this pool of Ypk1/2 (*Figure 7G*). In this model, loss of Ypk1/2 activity would cause rapid loss of Cln3 because Ypk1/2 directly promote production of Cln3. Inactivation of Sec7 would lead to accumulation of sphingolipids in the endoplasmic reticulum because they fail to be transported to the Golgi apparatus, which would inhibit the Ypk1/2-dependent increase in Cln3 during cell growth. Myriocin would lead to loss of inhibitory sphingolipids at the endoplasmic reticulum so that Ypk1/2 can drive an increase in Cln3, even though growth has stopped. Finally, addition of exogenous sphingolipids would lead to an increase in sphingolipid-dependent signals that inhibit Ypk1/2, leading to loss of Cln3. Several previous studies have suggested that Cln3 is localized to the endoplasmic reticulum and that regulation of Cln3 is closely associated with endoplasmic reticulum function (*Vergés et al., 2007*; *Yahya et al., 2014*).

Data from previous studies are consistent with a model in which there are multiple differentially regulated pools of TORC2 and Ypk1/2. For example, Ypk1/2 must carry out functions at the endoplasmic reticulum because the Orm1/2 proteins and ceramide synthase, two well-established direct targets of Ypk1/2, are localized there. Yet, TORC2 and Ypk1/2 are also found at the plasma membrane (*Berchtold and Walther, 2009*; *Niles et al., 2012*) and transport of sphingolipids or ceramide to the plasma membrane appears to be essential for relaying negative feedback signals that inhibit TORC2-dependent phosphorylation of Ypk1/2, consistent with the existence of signaling events at the plasma membrane (*Clarke et al., 2017*). Thus, there may be distinct, differentially regulated pools of Ypk1/2 at the endoplasmic reticulum and the plasma membrane.

At this point, numerous alternative models are possible, and our ability to construct detailed models is constrained by our limited understanding of the function and regulation of TORC2 and Ypk1/2. Nevertheless, the fact that Cln3, a critical dose-dependent regulator of cell size, is strongly influenced by signaling from key components of the TORC2 network suggests that a full understanding of cell size control will require a much better understanding of growth control and the TORC2 signaling network.

## Materials and methods
### Yeast strains, media, plasmids, and inhibitors
All strains are in the W303 background (*leu2-3, 112 ura3-1 can1-100 ade2-1 his3-11,15 trp1-1 GAL+ ssd1-d2*). All strains are derived from the parent strain DK186 with the exception of DK2961, which is derived from Y3507 (mat α) (gift from James R. Broach, Penn State). *Table 1* shows additional genetic features. One-step, PCR-based gene replacement was used for making deletions and adding epitope tags at the endogenous locus (*Longtine et al., 1998*; *Janke et al., 2004*). A plasmid carrying a copy of WHI5 was made by amplifying the *WHI5* gene with 329 bp upstream of the start codon and 159 bp downstream of the stop codon from genomic DNA with oligos *GCGGGATCCCCGTCTTCCTTGTGC TGTTTATG* and *GCGGAATTCGCGGATGTTGATCGGCGGAT*. The PCR fragment was cloned into the BamH1 and EcoR1 sites of YIplac211 to create plasmid pRAS1. To integrate an extra copy of WHI5 into the genome, the plasmid was cut with StuI to target integration at the *URA3* locus. The empty vector YIplac211 was integrated to create isogenic control strains.

Cells were grown in YP medium (1% yeast extract, 2% peptone) that contained 40 mg/L adenine and a carbon source, except where noted in the figure legends. Rich carbon medium (YPD) contained 2% dextrose, while poor carbon medium (YPG/E) contained 2% glycerol and 2% ethanol. In experiments using the ATP analog inhibitors 1-NM-PP1, 1-NA-PP1, or 3-MOB-PP1, no additional adenine was added to the media. CSM (MP Biosciences) contained either 2% dextrose (rich carbon) or 2% glycerol and 2% ethanol (poor carbon) with an additional 40 mg/L adenine. CSM without methionine was used for $^{35}$S-methionine labeling experiments.

Myriocin (Sigma) was solubilized in 100% methanol to make a 500 µg/mL stock solution. We have observed significant batch-to-batch differences in the effective concentration of myriocin from the same supplier. All ATP analog inhibitors were solubilized in 100% DMSO. 3-MOB-PP1 was a gift from Jack Stevenson and Kevan Shokat lab (UCSF). Auxin (indole-3-acetic acid) (Aldrich) stock was prepared at 50 mM in 100% EtOH and used at 1 mM.

### Cell size analysis and bud emergence
Cells were grown in YPD or YPG/E medium overnight at 22°C to mid-log phase (OD$_{600}$ less than 0.7). Cells were fixed with 3.7% formaldehyde for 30 min and were then washed with PBS + 0.02% Tween-20 + 0.1% sodium azide (PBTA) before measuring cell size using a Z2 Coulter Channelyzer and Z2 AccuComp v3.01a software as previously described (*Lucena et al., 2018*). For measurement of log phase cultures, each cell size plot is an average of at least three independent biological replicates in which each biological replicate is the average of three technical replicates. Biological replicates are measurements obtained from independent cultures grown on different days, whereas technical replicates are independent measurements of cultures grown on the same day. Between 30,000 and 40,000 cells are counted for each measurement. For *2xWHI5* measurements, three independent strain isolates generated by integration of the *WHI5* plasmid (pRAS1) were measured in more than three independent biological replicates and averaged. Similarly, the isogenic control strain

**Table 1.** Strains with multiple strain numbers indicate multiple independent isolates of the same strain.

| Strain | Genotype | Reference or source | Figures |
|---|---|---|---|
| DK186 | *MATa bar1* | **Altman and Kellogg, 1997** | **Figure 3** |
| DK3092 | *MATa bar1 WHI5-3xHA::HPHNT1 CLN3-6xHA::HIS3* | This study | **Figure 1 and Figure 1—figure supplement 1, Figure 1—figure supplement 2, Figure 2, Figure 3—figure supplement 1B** |
| DK4395 | *MATa bar1 WHI5-13MYC::HIS5* | This study | In support of **Figure 1**, data not shown |
| DK3669 DK3670 | *MATa bar1 URA3::YIplac211 (URA3)* | This study | **Figure 3A and B and Figure 3—figure supplement 1A** |
| DK3722 DK3875 DK3876 | *MATa bar1 URA3::pRAS1 (WHI5::URA3)* | This study | **Figure 3A and B and Figure 3—figure supplement 1A** |
| DK3726 DK3885 | *MATa bar1 whi5Δ::KANMX URA3::pRAS1 (WHI5::URA3)* | This study | **Figure 3—figure supplement 1A** |
| DK3728 | *MATa bar1 whi5Δ::KANMX URA3::YIplac211 (URA3)* | This study | **Figure 3—figure supplement 1A** |
| DK3716 DK3717 DK3718 | *MATa/α WHI5/whi5::KANMX* | This study | **Figure 3C** |
| DK3712 | *KANMX::TEF1-WHI5* | This study | **Figure 3D** |
| DK3323 | *KANMX::TEF1-WHI5-3xHA::HPHNT1 CLN3-6xHA::HIS3* | This study | **Figure 3—figure supplement 1B-D** |
| DK2017 | *MATa bar1 CLN3-6xHA::HIS3* | **Zapata et al., 2014** | **Figure 4A and B, Figure 5—figure supplement 1C and D, Figure 6D and E and Figure 6—figure supplement 1A-C, Figure 7A–F** |
| DK2907 | *MATa bar1 SEC7-V5-AID::KANMX CLN3-6xHA::NATNT2 pTIR2::HIS3 pTIR4::LEU2* | This study | **Figure 4C–F and Figure 4—figure supplement 1** |
| DK2936 | *MATa bar1 cdc28-as1(F88G) pho85-as1(F82G) CLN3-6xHA::HIS3* | This study | **Figure 5—figure supplement 1A and B** |
| DK2961 | *MATα sch9-as(T492G) CLN3-6xHA::NATNT2* | This study | **Figure 5** |
| DK3000 | *MATa bar1 ypk1-as (L424A) ypk2Δ::HIS3 CLN3-6xHA::NATNT2* | This study | **Figure 6A–C and Figure 6—figure supplement 1D and E** |
| DK3525 | *MATa bar1 ypk1Δ::HIS3 CLN3-6xHA::NATNT2* | This study | **Figure 6D** |
| DK3578 | *MATa bar1 lac1Δ::HIS3 lag1Δ::KANMX CLN3-6xHA::NATNT2* | This study | **Figure 7E and F** |

integrated with empty vector is an average of two independent isolates measured in more than three independent biological replicates and averaged. The percentage of budded cells was calculated by counting >200 cells at each time point using a Zeiss Axioskop 2 (Carl Zeiss).

## Centrifugal elutriation

Strains were grown in YPG/E medium overnight to OD$_{600}$ 0.4–0.8 at 30°C except for strains harboring temperature-sensitive alleles and the *ypk1-as ypk2Δ* strain, which were grown overnight at 22°C. Cells were harvested by spinning at 4000 rpm in a JLA 8.1 rotor at 4°C for 6 min. Cell pellets were resuspended in ~100 mL cold YPG/E, then sonicated for 1 min at duty cycle 0.5 using a Braun-Sonic U sonicator with a Braun 2000U probe at 4°C. Cells were loaded onto a Beckman JE-5.0 elutriator in a Beckman Coulter J6-MI centrifuge spinning at 2900 rpm at 4°C. After all cells were loaded, fluid flow was continued for 10 min to allow equilibration. Pump speed was then increased gradually to collect small unbudded cells, which were collected and pelleted by spinning in a JA-14 rotor at 5000 rpm for 5 min. Cells were resuspended into fresh medium at OD$_{600}$ 0.4–0.6 and grown at 25°C in a shaking water bath. For western blotting, 1.6 mL samples were collected at each time point as cells progressed through G1 phase. Since the cells do not divide during the time course, a constant number of cells is collected at each time point so that signals are intrinsically normalized on a per cell basis. Thus, western blot signals represent protein copy number per cell, rather than protein concentration. For Coulter counter analysis, samples were collected and fixed with 3.7% formaldehyde for 15–30 min for cell size measurements and to calculate the percentage of budded cells. Median cell size was calculated by the Coulter AccuComp software for each time point to plot cell size during growth in G1. We found that cell size measurements of *ypk1-as ypk2Δ* cells isolated from centrifugal elutriation were more consistent when cells were not fixed with formaldehyde but were measured immediately after washing once with PBTA to remove YP media.

## Western blotting

1.6 mL samples of cells taken from cultures were pelleted in a microfuge at 13,200 rpm for 15 s before aspirating the supernatant and adding 250 μL of glass beads and freezing on liquid nitrogen. Cells were lysed in 140 μL of 1× SDS-PAGE sample buffer (65 mM Tris-HCl, pH 6.8, 3% SDS, 10% glycerol, 100 mM β-glycerophosphate, 50 mM NaF, 5% β-mercaptoethanol, 2 mM PMSF, and bromophenol blue) by bead beating in a Biospec Mini-Beadbeater-16 at 4°C for 2 min. The lysate was centrifuged for 15 s to bring the sample to the bottom of the tube and was then incubated in a 100°C water bath for 5 min followed by centrifugation for 5 min at 13,200 rpm. Depending on the protein blot, 7.5–15 μL of the lysate were loaded into 10% acrylamide SDS-PAGE gels, then run at a constant current setting of 20 mA per gel at a maximum of 165 V. Gels were transferred to nitrocellulose membrane in a Bio-Rad Trans-Blot Turbo Transfer system. Blots were probed overnight at 4°C in 4% milk in western wash buffer (1× PBS + 250 mM NaCl +0.1% Tween-20) with mouse monoclonal anti-HA antibody (12CA5, gift of David Toczyski, University of California, San Francisco), polyclonal anti-Nap1 antibody (unpublished), polyclonal anti-Ypk1 antibody (*Alcaide-Gavilán et al., 2018*), or polyclonal rabbit anti-T662P antibody (gift from Ted Powers, University of California, Davis). Western blots using anti-T662P antibody were first blocked using TBST (10 mM Tris-Cl, pH 7.5, 100 mM NaCl, and 0.1% Tween-20) + 4% milk, followed by one wash with TBST, then incubated overnight with anti-T662P antibody in TBST +4% BSA. Western blots were incubated in secondary donkey anti-mouse (GE Healthcare NA934V) or donkey anti-rabbit (GE Healthcare NXA931 or Jackson Immunoresearch 711-035-152) antibody conjugated to HRP at room temperature for 60–90 min before imaging with Advansta ECL chemiluminescence reagents in a Bio-Rad ChemiDoc imaging system.

## Log phase nutrient shift

Cells were grown to mid-log phase (OD$_{600}$ 0.4–0.7) in rich carbon (YPD) at 22°C. Cells were shifted to poor carbon (YPG/E) using a vacuum filter apparatus with a 0.45 μM HAWP mixed cellulose ester filter. Cells were washed three times with room temperature YPG/E and then resuspended in room temperature YPG/E by pipetting and grown in a shaking water bath at 25°C.

## Western blot quantification

Western blots were quantified using Bio-Rad Imagelab software v6.0.1. For the elutriation experiments, relative signal was calculated as a ratio of the signal of each time point over the signal at either the zero time point (Whi5) or the 10 min time point (Cln3) (see figure legends for details). The signal was not normalized to the loading control because the loading control signal increased with growth during G1 phase. No Cln3 signal was detected at the 0 min time point in elutriated cells, so the 10 min time point was used as reference for that set of samples. Log phase samples were quantified by setting all signals relative to either the 0 min time point (*Figure 2*) or wild-type cells in rich carbon (*Figures 3B, 6D and 7E*). The signal was normalized to signal for the loading control band to correct for differences in total protein between samples.

## $^{35}$S-Methionine pulse labeling

Cells were grown overnight at room temperature to mid-log phase in CSM-methionine media containing 2% dextrose. The culture was concentrated to an $OD_{600}$ of 0.7 and incubated in a shaking water bath at 25°C. The vehicle or drug was added to the appropriate culture flask. A 1.2 mL sample of culture was transferred to a 1.6 mL screw top tube to label proteins made during the next 15 min. The labeling reaction was initiated by adding 1 μL of EasyTag L-[$^{35}$S]-Methionine from PerkinElmer at a stock concentration of 1 μCi/μL. The sample was mixed by vortexing and then placed into the 25°C shaking water bath. Labeling reactions were allowed to progress for 15 min, during which time samples were mixed by inversion every ~5 min. Labeling reactions were performed during 15 min intervals starting every 20 min. Samples were centrifuged at 13,200 rpm for 30 s, the supernatant was removed, and 250 μL of acid-washed beads were added before freezing on liquid nitrogen. Samples were prepared for SDS-PAGE similarly to samples described for western blotting in 140 μL of 1× sample buffer. After running the gel to the dye front, polyacrylamide gels were stained in R-250 Coomassie stain and then dried on Whatman paper using a Bio-Rad Gel Dryer. The dried gel was exposed film.

## Experimental replicates

All experiments were repeated for a minimum of three biological replicates, with the exception of *Figure 4—figure supplement 1D*, which is based on a single biological replicate. Biological replicates are defined as experiments carried out on different days with different starting cultures.

## Acknowledgements

We thank Andrew Murray, Jan Skotheim, Kurt Schmoller, Matthew Swaffer, Matthias Heinemann, Andreas Milias-Argeitis, and members of the lab for helpful discussions. We also thank Jack Stevenson and Kevan Shokat for providing 3-MOB-PP1. This work was funded by NIH grants GM053959 and GM131826.

## Additional information

### Funding

| Funder | Grant reference number | Author |
| --- | --- | --- |
| National Institutes of Health | GM053959 | Douglas R Kellogg |
| National Institutes of Health | GM131826 | Douglas R Kellogg |

The funders had no role in study design, data collection and interpretation, or the decision to submit the work for publication.

### Author contributions

Robert A Sommer, Formal analysis, Investigation, Methodology, Writing - original draft, Writing - review and editing; Jerry T DeWitt, Investigation, Methodology; Raymond Tan, Investigation; Douglas

R Kellogg, Conceptualization, Funding acquisition, Investigation, Methodology, Project administration, Writing - original draft, Writing - review and editing

### Author ORCIDs
Robert A Sommer  http://orcid.org/0000-0003-4529-0542
Jerry T DeWitt  http://orcid.org/0000-0002-9077-643X
Douglas R Kellogg  http://orcid.org/0000-0002-5050-2194

### Decision letter and Author response
Decision letter https://doi.org/10.7554/eLife.64364.sa1
Author response https://doi.org/10.7554/eLife.64364.sa2

## Additional files

### Supplementary files
• Transparent reporting form

### Data availability
Figures present data from biological replicates that are representative of multiple biological replicates. Coulter counter data show the average of multiple biological replicates.

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
