## [Decision Letter]

**Decision letter after peer review:**

Thank you for submitting your article "Growth-dependent signals drive an increase in early G1 cyclin concentration to link cell cycle entry with cell growth" for consideration by *eLife*. Your article has been reviewed by 3 peer reviewers, and the evaluation has been overseen by a Reviewing Editor and Naama Barkai as the Senior Editor. The reviewers have opted to remain anonymous.

Essential revisions:

As you will see below, all reviewers are highly supportive of the work, but asked for some clarification and some new data. In your revision, please address all points described below.

*Reviewer #1 (Recommendations for the authors):*

The experiments are performed to a high technical standard and in numerous biological replicates so the main conclusions are sound. My main concerns pertain to a few particular aspects of the results that are not explicitly discussed by the authors. While there are obviously many further experiments to be done to sort out the mechanism whereby Yck1/2 and sphingolipid metabolism control Cln3, these lie outside the reasonable scope of this study in my opinion. Nevertheless, some further discussion of possible mechanisms seems warranted. Finally, a few additional published observations should be mentioned and cited.

1. It is somewhat surprising that cells grown in glucose don't grow faster and that budding upon inoculation in rich is slower than in poor nutrients as a function of time. I would have expected the much faster growth rate in glucose to result in temporally faster budding than in glycerol/ethanol medium. What are the doubling times observed in rich versus poor medium?

2. It is also a bit puzzling that there is a transient burst of Cln3 immediately after inoculation into poor medium. This could be due to residual glucose in the yeast extract used to formulate YP medium, i.e., cells metabolize the small amount of glucose in the yeast extract before shifting to non-fermentative metabolism. This can be avoided by depleting YP of glucose by pre-culture with yeast in the absence of added carbon source (e.g., as a 5X YP stock, such that the yeast quickly consume all glucose then cease growth, the stock is then clarified and sterilized before use in subsequent experiments). Alternatively, to maintain complete nutrient homeostasis, cells can be elutriated in the clarified medium in which they were grown prior to elutriation and then diluted into the same medium. The transient burst doesn't affect the authors conclusions but a similar burst has been observed in other studies where residual glucose was not properly accounted for. The authors may wish to mention this.

3. It appears that the Cln3 signal reaches a maximum well before Start (e.g., 30-50 min before 20% budding is reached in most experiments) but the authors do not comment on this finding. This result aligns with recent single cell analysis from the Heinemann lab, which also suggests that Cln3 peaks well before Start (Litsios et al., 2019). It would be worth addressing this point in the discussion, including the caveat that population level studies can obscure single cell effects.

4. Why is bulk protein synthesis is not affected in the SEC7-AID strain despite the complete block to growth and what does this mean? Is the protein concentration going up in the arrested G1 phase cells? Or is there a higher rate of protein degradation to compensate for on-going synthesis? It does appear that the loading control increases throughout the experiment consistent with the former possibility. It would be useful for the authors to show the 35S incorporation data for the elutriated cultures so that the effect in G1 phase can be assessed (this is currently cited as data not shown). If the protein concentration in the arrested cells is higher, do the arrested cells rapidly grow and pass Start quickly upon release from the AID block?

5. A more precise description of key technical details would help the reader understand the exact experimental set-up and results.

(a) How were cell volumes estimated? half peak height on the daughter size of the size plot or mode size?

(b) For all elutriation experiments, the volume of each fraction and the percent budding should be indicated above each lane so that readers can estimate the time of Start. For at least one elutriation experiment, representative size plots and FACS profiles of the starting size fraction and progressive timepoints should be shown to indicate uniformity of the starting distribution and synchronous progression through the cell cycle.

(c) If I interpret the methods correctly, each elutriation-release timepoint represents a fixed volume of cells and samples are not normalized for total protein content prior to loading on gels. This should be stated explicitly when the elutriation experiments are first described so that readers understand that the signals are intrinsically normalized on a per cell basis (for cells still in G1 phase), i.e., the signal on the blots represents copy number, not concentration. Also, what fraction of each 1.6 mL timepoint was loaded on the gels?

6. The main unanswered question is the mechanism whereby Cln3 accumulation is prevented by sphingolipid accumulation. If, for example, Cln3 translation is dependent on Ypk1/2 activity, might this be mediated by uORFs in the CLN3 mRNA? Testing effects on a version of CLN3 that lacks the uORFs would answer this question. Or is Cln3 itself a substrate for Ypk1/2, perhaps stabilizing Cln3 to allow its accumulation? Or is there an effect on CLN3 mRNA levels? Also, does the feedback inhibition from sphingosine affect TORC2 activity, which in turn may affect Cln3? Does a reduction in TORC2 activity affect Cln3 accumulation? While these experiments are probably beyond the scope of the study, some discussion is warranted.

7. The results of this study underscore the major question of how cells in poor nutrients with low levels of Cln3 (and also Cln1/2, shown by Dorsey et al., 2018) activate Start at a smaller cell size, i.e., despite low Cln-Cdc28 activity. One possible contribution is the higher levels of G1/S transcription factors (Swi4, Swi6, Mbp1) in poor nutrients described by Dorsey et al., (2018), particularly for Mbp1, which is not inhibited by Whi5. This result should be cited along with the observation from Dorsey et al., that G1/S TF abundance is limiting with respect to promoter/TF binding site copy number in early G1 phase cells. Also, on a note for citation completeness, Dorsey et al., should be cited in the sentence "…consistent with several previous studies that used fluorescence microscopy to analyze Whi5 levels (Schmoller et al., 2015; Litsios et al., 2019)." on p5.

8. Other literature considerations:

(a) Importantly, the authors point out that some of the key experiments in the Schmoller paper were obtained with a bck2 strain that is defective for Start (which they replicate) and then show these effects do not hold in wild type cells. Costanzo et al., (2004) reported that overexpression of GAL1-WHI5 in a cln3 strain causes a G1 arrest, and that overexpression in a wild type strain causes a G1 delay. Similarly, overexpression of GAL1-WHI5 causes a severe growth defect in cln1 cln2 strain background. In addition, Costanzo et al., found that GAL1-WHI5 had only moderate effects on size and cell cycle distribution in wild type cells. These results are consistent with the authors' WHI5 dosage results and should be cited appropriately.

(b) The proposed links between the secretory system and size are intriguing. In this regard, it is worth noting that two previous studies uncovered a direct connection between the secretory system, ribosome biogenesis and cell size (Lempiäinen et al., 2009; Singh and Tyers 2009). The authors cite previous studies purporting ER localization of Cln3 as a possible mechanism for sphingolipid effects but this data should be interpreted cautiously given the difficulty in detecting low abundance Cln3 signals at the single cell level.

(c) A key previous result often overlooked in nutrient shift experiments is that rates of rRNA transcription reach maximal levels within a few minutes of shift from ethanol to glucose medium whereas protein synthetic rate does not achieve a maximum until nearly an hour after the shift (Kief and Warner 1981). Because cells adjust their critical cell-size threshold very rapidly upon nutrient shifts (Lorincz and Carter 1979), this result suggests that the rate of ribosome biogenesis and not protein synthesis per se sets the size threshold. The work of Keif and Warner could be cited in the sentences "In both budding yeast and fission yeast, cells rapidly readjust the threshold amount of growth required for cell cycle progression when shifted to nutrient conditions that support different growth rates (Fantes and Nurse 1977; Johnston et al., 1979; Lucena et al., 2018). In fission yeast, the threshold appears to be readjusted within minutes." on p8.

(d) In addition to previous single cell studies on Whi5 abundance as a function of size using live cell imaging in either time course studies or fixed single images, a recent publication using PALM imaging also showed that Whi5 is not diluted as a function of size (Black et al., 2019).

*Reviewer #2 (Recommendations for the authors):*

My major recommendations are included in the public review. I would be particularly interested to see the authors test the bck2∆ strain from the Schmoller paper with by western blot experiments as in Figure 1.

*Reviewer #3 (Recommendations for the authors):*

1. Cln3 was made detectable on western blots by tagging with 6xHA. It is crucial to demonstrate that tinkering with the 3'UTR of CLN3 does not lead to changes in Cln3 levels or activity. Indeed, it is published that a tagged version of Cln3 (CLN3-PrA) exhibits about a 10% increase in cell volume (Cross et al., 2002). This can be done easily by comparing cell size distribution profiles of WT and CLN3-6xHA cells, in rich and poor medium, as done for other constructs in Figure 3A, 3C and Figure 3_SF1. The same should be done with Whi5-3xHA.

2. The most dramatic increase in Cln3 levels occurs when G1 cells previously grown in poor carbon medium are transferred to rich carbon source (dextrose). It is well established that a sudden rewiring of carbon metabolism ('sugar shock') delays CLN1,2 transcription and requires higher Cln3 accumulation (e.g. Baroni et al., 1994; Tokiwa et al., 1994; Flick et al., 1998). It is important to determine how much Cln3 has to accumulate before Start occurs when elutriated small G1 cells pre-grown in rich medium are transferred to the same rich medium.

3. Figure 5 seems to rule out that cell size at Start is controlled by the Sch9 kinase regulated by TORC1, which is at odds with previous reports showing reduced cell growth and smaller size of sch9∆ cells. What is the rationale of adding 1-NM-PP1 45 minutes after inoculation of small G1 cells in the medium and not earlier. Moreover, the experiment in Figure 5 does not control for Sch9-as inactivation by 1-NM-PP1. This could be done by looking at phosphorylation of the downstream target Rsp6 (González et al., 2015). As the TOR1/Sch9 pathway promotes protein synthesis (Eltschinger and Loewith, 2016), it is not surprising that inhibition, even partial, of Sch9 leads to a decrease in Cln3 levels. It is important that the authors show the status of the loading control that they use throughout the manuscript and quantify the level of Cln3 as it is performed on figure 1 for instance. It is also important to control that 1-NM-PP1 has no off-target effects on growth or budding by monitoring its effect on WT cells (as done for 3-MOB in Figure 6 SF1). As it is, the experiment is unconclusive and should be put in SupFigure at best.

4. Figure 6E is used to indicate that Ypk1 phosphorylation by TORC2 might favour Cln3 accumulation. This conclusion seems premature as slow-migrating Ypk1 bands appear at the time (10 min) when Cln3 also up-shifts and disappears, suggesting in fact that Ypk1 phosphorylation might correlate with Cln3 proteolysis. The same figure 6E is described in two paragraphs (on page 11 and 12) that should be merged. It is puzzling to read that the conclusions written in each paragraph differ from one another. In the first one the authors claim that 'A reappearance of Cln3 at the end of the time course was accompanied by another decrease in Ypk1 phosphorylation'. In the second one, they claim that 'The increase in Cln3 that occurred later in the time course as cells adapted to the new carbon source was correlated with an increase in TORC2-dependent phosphorylation of Ypk1/2.' The latter seems to be supported by the WB displayed. Nevertheless, a quantification of Cln3 level, Ypk1 phosphorylation level along with the level of the loading control would be helpful to support the claim. To this reviewer knowledge, Cln3 ubiquitin-dependent proteolysis is triggered by Cdk1 mediated phosphorylation. The positive arrow connecting Ypk1,2 and Cln3 in the model presented in Figure 7 is not sustained conclusively by the data.

Comments and suggestions

1. There is lots of emphasis on Cln3 and Whi5 levels, but what counts for Start is the phosphorylation status or nuclear exclusion of Whi5. Whi5 appears as a doublet on some western blots (e.g. Figure 2, Figure 3B,D), a putative proxy for Whi5 phosphorylation. This could be used to determine which other kinases or phosphatases control Whi5's phosphorylation status when Cln3 levels are very low in cells grown on poor carbon medium.

2. It is said at several places in the manuscript that Cln3 accumulates gradually during G1. This may be true when looking at large cell populations, but Litsios et al., have shown that Cln3 is synthesized in bursts during G1, following metabolic oscillations. These bursts do not take place at exactly the same time/size in all cells, but may translate as a gradual/linear increase when looking at the population of cells. This could be worth mentioning in the manuscript.

3. In all but one experiment (Figure 3 —figure supplement 1B), cells were grown in a rich-nutrient medium (YP, 1% yeast, 2% peptone) that contained 2 types of carbon sources, either dextrose or 2% glycerol/2% ethanol referred to as rich and poor carbon medium, respectively. In the experiment displayed on Figure 3 —figure supplement 1B, cells were grown in a poor-nutrient synthetic complete medium. However, the authors mix rich or poor carbon sources with rich or poor nutrients. For the sake of clarity, they should use rich or poor carbon on all figures panels and keep rich or poor nutrients only for Figure 3 —figure supplement 1B.

4. Based on their data the authors argue that it is the increase of Cln3 concentration rather than the dilution of Whi5 that would control cell cycle entry. The key quantification of this is shown in Figure 1 SF1C,D and should be moved to the main panel.

5. p17-18: To support the claim that accumulation of sphingolipid would inhibit the Ypk1/2-dependent increase in Cln3, the authors could monitor the phosphorylation status of Ypk1/2 in sec7 mutants and/or in cells incubated with phytosphingosine. Alternatively, they could analyze whether addition of phytosphingosine on ypk1∆ or ypk1-as ypk2∆ mutant leads to a further decrease of Cln3 level or not.

Data presentation

1. Figure 1E: It should be mentioned that it represents the quantification of the WB displayed in D, and if the signal was normalized to a loading control. Please explain why this was not done if it was not.

2. Figure 3 and figure 3 —figure supplement 1: Two different genotypes are used 2xWHI5 and Wild type + WHI5. It is not clear whether there is a difference between them. If there is none, the use of only 1 genotype would help for the comprehension. It there is a difference, it should be explained.

3. Figure 3D: Contrary to the authors sentence (p9 3rd paragraph) Cln3 level is not displayed on the panel. If it is relevant, it should be shown. If not, the sentence should be changed. The authors should add a loading control for this panel, which is not shown in figure 3 —figure supplement 1E as stated in the legend of the figure.

4. Figure 5: the panel of the loading control is missing in the figure, while it is mentioned in the legend. It should be added.

5. Figure 6 —figure supplement 1B is not cited in the text. A loading control should be added to the western blot.

6. p12, 4th paragraph; figure 7C and 7D are not cited accurately in the text.

7. Table 1: Some strains are not referenced correctly to the corresponding figures. For DK3572, it should be S3C, for DK3743, it should be S3D. The strain 2xWHI5 used in figure 3 —figure supplement 1 is not referenced in the table 1.

8. Overall, the manuscript is quite long with some repeats, and would benefit from some shortening.

– Figure 4 A,B and the figure 5 —figure supplement 1 could be merged in a unique figure and the corresponding text simplified, as the conclusions drawn from these results are similar.

– p12, 3rd paragraph; the first and third sentence are redundant. It should be rephrased.

– The Discussion could be shortened as results or interpretations already presented in Results are merely rephrased, for instance p15 1st paragraph, p16 4th and 5th paragraphs…

---

## [Author Response]

Essential revisions:As you will see below, all reviewers are highly supportive of the work, but asked for some clarification and some new data. In your revision, please address all points described below.Reviewer #1 (Recommendations for the authors):The experiments are performed to a high technical standard and in numerous biological replicates so the main conclusions are sound. My main concerns pertain to a few particular aspects of the results that are not explicitly discussed by the authors. While there are obviously many further experiments to be done to sort out the mechanism whereby Yck1/2 and sphingolipid metabolism control Cln3, these lie outside the reasonable scope of this study in my opinion. Nevertheless, some further discussion of possible mechanisms seems warranted. Finally, a few additional published observations should be mentioned and cited.1. It is somewhat surprising that cells grown in glucose don't grow faster and that budding upon inoculation in rich is slower than in poor nutrients as a function of time. I would have expected the much faster growth rate in glucose to result in temporally faster budding than in glycerol/ethanol medium. What are the doubling times observed in rich versus poor medium?

For the experiment shown in Figure 1, cells were first grown overnight in YP media containing a poor carbon source (2% glycerol/2% ethanol). Under these conditions, cells are born at a very small size and spend a longer time growing in G1 phase, which facilitates isolation of a uniform population of small unbudded cells in early G1 phase. After isolation of small unbudded cells by elutriation, the cells were divided and half were grown in the same media, while the other half were shifted to a rich carbon source (2% glucose). The cells shifted to rich carbon grow faster but they also immediately increase the growth requirement for cell cycle entry and therefore must undergo an unusually long G1 phase to reach the larger cell size required for cell cycle entry in rich carbon. In contrast, the cells that remain in poor carbon can undergo cell cycle entry at a much smaller size and therefore spend less time in G1 phase. We have edited the text to highlight these issues and how the influence interpretation of the results. See also our response to Reviewer 2.

In a previous publication, we provided detailed information regarding growth rates during each phase of the cell cycle in cells grown continuously in rich or poor carbon (PMID: 28939614). This showed that growth rate varies during the cell cycle, and that cells in rich carbon grow 2-3 fold faster than cells in poor carbon.

2. It is also a bit puzzling that there is a transient burst of Cln3 immediately after inoculation into poor medium. This could be due to residual glucose in the yeast extract used to formulate YP medium, i.e., cells metabolize the small amount of glucose in the yeast extract before shifting to non-fermentative metabolism. This can be avoided by depleting YP of glucose by pre-culture with yeast in the absence of added carbon source (e.g., as a 5X YP stock, such that the yeast quickly consume all glucose then cease growth, the stock is then clarified and sterilized before use in subsequent experiments). Alternatively, to maintain complete nutrient homeostasis, cells can be elutriated in the clarified medium in which they were grown prior to elutriation and then diluted into the same medium. The transient burst doesn't affect the authors conclusions but a similar burst has been observed in other studies where residual glucose was not properly accounted for. The authors may wish to mention this.

In Figure 2, the cells are shifted from a very rich carbon medium (YP + 2% glucose) into a very poor carbon medium (YP + 2% glycerol/ethanol). We are careful to ensure that the cells pre-grown in rich glucose media are kept at low density so that they have not significantly depleted the glucose. Therefore, even if there is residual glucose in the yeast extract used to formulate YP medium, the concentration of that residual glucose must be far lower than the 2% glucose in the rich carbon medium so that the cells experience a dramatic reduction in the availability of glucose. We therefore think that it is unlikely that the transient burst in Cln3 is due to residual glucose, but we would be happy to mention this possibility if requested by the reviewer.

3. It appears that the Cln3 signal reaches a maximum well before Start (e.g., 30-50 min before 20% budding is reached in most experiments) but the authors do not comment on this finding. This result aligns with recent single cell analysis from the Heinemann lab, which also suggests that Cln3 peaks well before Start (Litsios et al., 2019). It would be worth addressing this point in the discussion, including the caveat that population level studies can obscure single cell effects.

We agree and have added a reference to the Litsios et al., results in our discussion of the data shown in Figure 1. We have also included new text to highlight the fact that population-level analysis can miss effects seen in single cells.

4. Why is bulk protein synthesis is not affected in the SEC7-AID strain despite the complete block to growth and what does this mean? Is the protein concentration going up in the arrested G1 phase cells? Or is there a higher rate of protein degradation to compensate for on-going synthesis? It does appear that the loading control increases throughout the experiment consistent with the former possibility. It would be useful for the authors to show the 35S incorporation data for the elutriated cultures so that the effect in G1 phase can be assessed (this is currently cited as data not shown). If the protein concentration in the arrested cells is higher, do the arrested cells rapidly grow and pass Start quickly upon release from the AID block?

Classic work carried out by the Schekman lab found that protein production continues after an arrest of membrane traffic, and the resulting increase in cell density was a key tool in the genetic screens that led to isolation of the sec mutants. The western blot data in Figure 4E and the S^35^ incorporation data for log phase cells shown in Figure 4F are consistent with the Schekman results. We have included S^35^ incorporation data for the synchronized sec7-AID cells in Figure 4 —figure supplement 1D, which are also consistent with the Schekman results.

The idea that accumulation of extra protein in growth-arrested sec7-AID cells could lead to faster growth and cell cycle entry upon release from the arrest is intriguing. However, we have not yet tested this due to concerns that we would lack a suitable reference control that would allow us to determine if the cells pass Start more quickly. There is also a concern that the sec7-AID cells could require extra time to recover from the disruption of membrane traffic.

5. A more precise description of key technical details would help the reader understand the exact experimental set-up and results.(a) How were cell volumes estimated? half peak height on the daughter size of the size plot or mode size?

The reported cell volumes are median cell size. This information has been included in the Methods section.

(b) For all elutriation experiments, the volume of each fraction and the percent budding should be indicated above each lane so that readers can estimate the time of Start. For at least one elutriation experiment, representative size plots and FACS profiles of the starting size fraction and progressive timepoints should be shown to indicate uniformity of the starting distribution and synchronous progression through the cell cycle.

We have added a new supplemental figure that shows Coulter Counter plots for successive time points for the elutriation experiment in Figure 1, which shows a tight starting distribution that increases in size with time (Figure 1 —figure supplement 1A).

Space is very tight in Figure 1 so we found that we were not able to add volumes and percent budding above each lane without disrupting the layout of the figure. Adding the numbers also made the panel look more crowded, and it seemed to repeat data in the other plots showing percent bud emergence and change in volume. We would therefore prefer to keep the data about percent budding and volume in separate plots, but would be happy to try to rearrange the data if requested by the reviewer.

(c) If I interpret the methods correctly, each elutriation-release timepoint represents a fixed volume of cells and samples are not normalized for total protein content prior to loading on gels. This should be stated explicitly when the elutriation experiments are first described so that readers understand that the signals are intrinsically normalized on a per cell basis (for cells still in G1 phase), i.e., the signal on the blots represents copy number, not concentration. Also, what fraction of each 1.6 mL timepoint was loaded on the gels?

This is a great suggestion that would help clarify the manuscript. We have added information to the main text to explain that a constant number of cells is collected for each time point. For each time point, cells are lysed into 140 ul of sample buffer and 15 ul are loaded on the gel.

6. The main unanswered question is the mechanism whereby Cln3 accumulation is prevented by sphingolipid accumulation. If, for example, Cln3 translation is dependent on Ypk1/2 activity, might this be mediated by uORFs in the CLN3 mRNA? Testing effects on a version of CLN3 that lacks the uORFs would answer this question. Or is Cln3 itself a substrate for Ypk1/2, perhaps stabilizing Cln3 to allow its accumulation? Or is there an effect on CLN3 mRNA levels? Also, does the feedback inhibition from sphingosine affect TORC2 activity, which in turn may affect Cln3? Does a reduction in TORC2 activity affect Cln3 accumulation? While these experiments are probably beyond the scope of the study, some discussion is warranted.

We agree that these are amongst the most important next questions and that they are beyond the scope of the current study. We have added new text to highlight these questions.

7. The results of this study underscore the major question of how cells in poor nutrients with low levels of Cln3 (and also Cln1/2, shown by Dorsey et al., 2018) activate Start at a smaller cell size, i.e., despite low Cln-Cdc28 activity. One possible contribution is the higher levels of G1/S transcription factors (Swi4, Swi6, Mbp1) in poor nutrients described by Dorsey et al., (2018), particularly for Mbp1, which is not inhibited by Whi5. This result should be cited along with the observation from Dorsey et al., that G1/S TF abundance is limiting with respect to promoter/TF binding site copy number in early G1 phase cells. Also, on a minor note for citation completeness, Dorsey et al., should be cited in the sentence "…consistent with several previous studies that used fluorescence microscopy to analyze Whi5 levels (Schmoller et al., 2015; Litsios et al., 2019)." on p5.

We have added new text to cite the results and model of Dorsey et al., We have also added the missing Dorsey et al., citation.

8. Other literature considerations:(a) Importantly, the authors point out that some of the key experiments in the Schmoller paper were obtained with a bck2 strain that is defective for Start (which they replicate) and then show these effects do not hold in wild type cells. Costanzo et al., (2004) reported that overexpression of GAL1-WHI5 in a cln3 strain causes a G1 arrest, and that overexpression in a wild type strain causes a G1 delay. Similarly, overexpression of GAL1-WHI5 causes a severe growth defect in cln1 cln2 strain background. In addition, Costanzo et al., found that GAL1-WHI5 had only moderate effects on size and cell cycle distribution in wild type cells. These results are consistent with the authors' WHI5 dosage results and should be cited appropriately.

We agree and have included the citation for the Costanzo data showing that GAL1-WHI5 has modest effects. We were uncertain how the GAL1-WHI5 expression data in cln3∆ and cln1/2∆ cells is relevant in the context of the dilution model and therefore have not mentioned those results, but please let us know if we are missing something.

(b) The proposed links between the secretory system and size are intriguing. In this regard, it is worth noting that two previous studies uncovered a direct connection between the secretory system, ribosome biogenesis and cell size (Lempiäinen et al., 2009; Singh and Tyers 2009). The authors cite previous studies purporting ER localization of Cln3 as a possible mechanism for sphingolipid effects but this data should be interpreted cautiously given the difficulty in detecting low abundance Cln3 signals at the single cell level.

We agree and have added the additional citations. We were unaware of these papers and found them to be quite interesting.

(c) A key previous result often overlooked in nutrient shift experiments is that rates of rRNA transcription reach maximal levels within a few minutes of shift from ethanol to glucose medium whereas protein synthetic rate does not achieve a maximum until nearly an hour after the shift (Kief and Warner 1981). Because cells adjust their critical cell-size threshold very rapidly upon nutrient shifts (Lorincz and Carter 1979), this result suggests that the rate of ribosome biogenesis and not protein synthesis per se sets the size threshold. The work of Keif and Warner could be cited in the sentences "In both budding yeast and fission yeast, cells rapidly readjust the threshold amount of growth required for cell cycle progression when shifted to nutrient conditions that support different growth rates (Fantes and Nurse 1977; Johnston et al., 1979; Lucena et al., 2018). In fission yeast, the threshold appears to be readjusted within minutes." on p8.

The Johnston et al., 1979 article that we cited in the original version includes Lorincz and Carter, as well as the data showing that critical cell size is rapidly reset after a nutrient shift. We were not able to find another Lorincz and Carter article on PubMed, so we believe that the Johnston article is the relevant article, but please let us know if we missed an article. We have also included a citation to the Kief and Warner article.

(d) In addition to previous single cell studies on Whi5 abundance as a function of size using live cell imaging in either time course studies or fixed single images, a recent publication using PALM imaging also showed that Whi5 is not diluted as a function of size (Black et al., 2019).

We agree and have added the Black et al., citation in several places in the manuscript.

Reviewer #2 (Recommendations for the authors):My major recommendations are included in the public review. I would be particularly interested to see the authors test the bck2∆ strain from the Schmoller paper with by western blot experiments as in Figure 1.

We are curious about this too, and there are a number of other experiments that we have considered carrying out to reconcile our results with the Schmoller et al., results. However, we’ve had our hands full with the other experiments so we found it necessary to put these experiments on hold. In addition, we have wanted to focus as much as possible on how cell size control works in wild type cells, rather than in mutant backgrounds.

Reviewer #3 (Recommendations for the authors):1. Cln3 was made detectable on western blots by tagging with 6xHA. It is crucial to demonstrate that tinkering with the 3'UTR of CLN3 does not lead to changes in Cln3 levels or activity. Indeed, it is published that a tagged version of Cln3 (CLN3-PrA) exhibits about a 10% increase in cell volume (Cross et al., 2002). This can be done easily by comparing cell size distribution profiles of WT and CLN3-6xHA cells, in rich and poor medium, as done for other constructs in Figure 3A, 3C and Figure 3_SF1. The same should be done with Whi5-3xHA.

We agree that this is a good test and the data are shown in a new figure panel (Figure 1 —figure supplement 1B). We found that Cln3-6XHA causes a slight increase in cell size in YP medium containing rich carbon, as observed previously for Cln3 tagged with protein A (Cross, 2002). Cln3-6XHA caused little effect on cell size in YP medium containing poor carbon. A previous study found that Cln3 tagged with protein A rescues a complete loss of function of CLN3 in genetic backgrounds that require CLN3 function, which suggests that tagged versions of Cln3 retain most of the normal functions of Cln3 (Cross, 2002). It would be ideal to have an antibody that recognizes Cln3. We have successfully raised antibodies that recognize over 25 proteins in our lab, but Cln3 has been uniquely difficult. We attempted to express multiple regions of the protein in *E. coli*, but in every case the proteins have been completely insoluble and refractory to purification. We are still working on making a Cln3 antibody and our current approach is to purify large amounts of stabilized Cln3 from yeast using methods similar to those used by the Diffley lab to purify DNA replication proteins. However, even in a best case scenario an antibody is at least 6 months away and the approach may not work.

Previous studies found that fluorescently tagged versions of Whi5 do not cause a decrease in cell size, which suggested that tagged versions of Whi5 are fully functional. We found that Whi5-3XHA cells are slightly smaller than wild type cells (Figure 1 —figure supplement 1B). We also found that Whi5-13Myc tag does not cause a decrease in cell size, which suggests that the HA tag sequence has a mild effect on Whi5 function. We further found that Whi5-3XHA and Whi5-13Myc showed the same behavior in elutriated cells undergoing growth in G1 phase. We therefore used the Whi5-3XHA so that the behavior of Cln3-6XHA and Whi5-3XHA could be directly compared in the same western blot samples. We have included new text in the Results section describing the properties of the Cln3 and Whi5 fusion proteins. In addition, we repeated the TEF1-WHI5 expression experiments in Figure 3 using untagged versions of WHI5.

This important issue was a blind spot for us in the original version, so we thank the reviewer for pointing it out.

2. The most dramatic increase in Cln3 levels occurs when G1 cells previously grown in poor carbon medium are transferred to rich carbon source (dextrose). It is well established that a sudden rewiring of carbon metabolism ('sugar shock') delays CLN1,2 transcription and requires higher Cln3 accumulation (e.g. Baroni et al., 1994; Tokiwa et al., 1994; Flick et al., 1998). It is important to determine how much Cln3 has to accumulate before Start occurs when elutriated small G1 cells pre-grown in rich medium are transferred to the same rich medium.

We attempted to grow cells in YPD, isolate small unbudded cells, and then release the cells back into YPD. However, we found that it was not possible to isolate a uniform population of small unbudded cells under these conditions. The problem is that very little growth occurs in G1 phase in YPD so that newly born cells are nearly the same size as mother cells (PMID: 28939614). This, combined with the normal variation in cell size observed in wild type yeast, means that elutriation yields a mix of unbudded and budded cells. Others have faced the same problem (PMID: 31685990, 10728640). The fact that so little growth occurs in G1 phase in YPD is an additional argument against the idea that dilution of Whi5 plays a substantial and general role in cell size control.

As an alternative, we grew cells in complete synthetic medium (CSM) containing 2% glucose. Under these conditions, cells grow more slowly and are smaller because CSM is limiting for nutrients other than glucose. We isolated small unbudded cells and released them into the same medium so that there would not be shift in carbon source. We found that Cln3 levels increase 3-fold, while Whi5 levels do no change substantially, similar to the results for cells growing in YP medium containing poor carbon. These data are shown in a new figure (Figure 1 —figure supplement 2).

3. Figure 5 seems to rule out that cell size at Start is controlled by the Sch9 kinase regulated by TORC1, which is at odds with previous reports showing reduced cell growth and smaller size of sch9∆ cells. What is the rationale of adding 1-NM-PP1 45 minutes after inoculation of small G1 cells in the medium and not earlier. Moreover, the experiment in Figure 5 does not control for Sch9-as inactivation by 1-NM-PP1. This could be done by looking at phosphorylation of the downstream target Rsp6 (González et al., 2015). As the TOR1/Sch9 pathway promotes protein synthesis (Eltschinger and Loewith, 2016), it is not surprising that inhibition, even partial, of Sch9 leads to a decrease in Cln3 levels. It is important that the authors show the status of the loading control that they use throughout the manuscript and quantify the level of Cln3 as it is performed on figure 1 for instance. It is also important to control that 1-NM-PP1 has no off-target effects on growth or budding by monitoring its effect on WT cells (as done for 3-MOB in Figure 6 SF1). As it is, the experiment is unconclusive and should be put in SupFigure at best.

The logic for adding inhibitor at 45 minutes was that we wanted to see whether inhibition of sch9 caused rapid loss of Cln3 levels, as we observed for inhibition of ypk1-as. We were also concerned that adding inhibitor earlier would stop the cell cycle before Cln3 accumulation had been initiated.

The sch9-as allele was first reported by Jorgensen et al., 2004, who showed that sch9-as cells are smaller than wild type cells in the absence of inhibitor, consistent with the fact that most analog-sensitive kinases have partially compromised function. Addition of the inhibitor made the cells very small, consistent with complete or nearly complete inhibition of sch9-as kinase activity. We obtained the same results for our sch9-as strain, which verified the version of the strain in our collection (see below). The fact that addition of inhibitor causes a transient loss of Cln3 in sch9-as cells provides further evidence that the inhibitor is working.

We have included the loading control for the sch9-as experiment and quantified Cln3 levels (Figures 5C and Figure 5 —figure supplement 1D). Our data are not necessarily at odds with the previous sch9∆ data. Both sch9∆ and prolonged inhibition of sch9-as cause large reductions in growth rate and cell size. Our data suggest that in both cases the reductions could be a consequence of long term effects on the rate of ribosome biogenesis.

4. Figure 6E is used to indicate that Ypk1 phosphorylation by TORC2 might favour Cln3 accumulation. This conclusion seems premature as slow-migrating Ypk1 bands appear at the time (10 min) when Cln3 also up-shifts and disappears, suggesting in fact that Ypk1 phosphorylation might correlate with Cln3 proteolysis. The same figure 6E is described in two paragraphs (on page 11 and 12) that should be merged. It is puzzling to read that the conclusions written in each paragraph differ from one another. In the first one the authors claim that 'A reappearance of Cln3 at the end of the time course was accompanied by another decrease in Ypk1 phosphorylation'. In the second one, they claim that 'The increase in Cln3 that occurred later in the time course as cells adapted to the new carbon source was correlated with an increase in TORC2-dependent phosphorylation of Ypk1/2.' The latter seems to be supported by the WB displayed. Nevertheless, a quantification of Cln3 level, Ypk1 phosphorylation level along with the level of the loading control would be helpful to support the claim. To this reviewer knowledge, Cln3 ubiquitin-dependent proteolysis is triggered by Cdk1 mediated phosphorylation. The positive arrow connecting Ypk1,2 and Cln3 in the model presented in Figure 7 is not sustained conclusively by the data.

We agree that our interpretation of the data in Figure 6E were confusing in the original version. Part of the confusion may arise from a lack of clarity in our writing and in the literature about the different phosphorylation inputs into Ypk1/2. The literature suggests that changes in the electrophoretic mobility of Ypk1 could be due largely to the Fpk1/2 kinases. TORC2 also influences Ypk1/2 phosphorylation, as detected by a phosphospecific antibody, but it remains unclear whether TORC2 also influences the electrophoretic mobility of Ypk1/2. The data suggest phosphorylation of Ypk1/2 that can be detected via electrophoretic mobility shifts is correlated with Cln3 levels, while TORC2-dependent phosphorylation with a phosphospecific antibody is not well correlated with Cln3 levels. We have edited the manuscript to make this more clear and to clarify what can and cannot be concluded from the data.

After editing this section for greater clarity, it seems to make sense to keep the two paragraphs separate to emphasize that electrophoretic mobility shifts of Ypk1/2 and the phosphospecific antibody are likely measuring different inputs into Ypk1/2.

Comments and suggestions1. There is lots of emphasis on Cln3 and Whi5 levels, but what counts for Start is the phosphorylation status or nuclear exclusion of Whi5. Whi5 appears as a doublet on some western blots (e.g. Figure 2, Figure 3B,D), a putative proxy for Whi5 phosphorylation. This could be used to determine which other kinases or phosphatases control Whi5's phosphorylation status when Cln3 levels are very low in cells grown on poor carbon medium.

We agree, but feel that further experiments along these lines are beyond the scope of this study.

2. It is said at several places in the manuscript that Cln3 accumulates gradually during G1. This may be true when looking at large cell populations, but Litsios et al., have shown that Cln3 is synthesized in bursts during G1, following metabolic oscillations. These bursts do not take place at exactly the same time/size in all cells, but may translate as a gradual/linear increase when looking at the population of cells. This could be worth mentioning in the manuscript.

We agree and have edited the manuscript to mention that our population-level studies could miss the transient bursts in translation detected by Litsios.

3. In all but one experiment (Figure 3 —figure supplement 1B), cells were grown in a rich-nutrient medium (YP, 1% yeast, 2% peptone) that contained 2 types of carbon sources, either dextrose or 2% glycerol/2% ethanol referred to as rich and poor carbon medium, respectively. In the experiment displayed on Figure 3 —figure supplement 1B, cells were grown in a poor-nutrient synthetic complete medium. However, the authors mix rich or poor carbon sources with rich or poor nutrients. For the sake of clarity, they should use rich or poor carbon on all figures panels and keep rich or poor nutrients only for Figure 3 —figure supplement 1B.

We agree and have edited to clarify which media are used in each experiment.

4. Based on their data the authors argue that it is the increase of Cln3 concentration rather than the dilution of Whi5 that would control cell cycle entry. The key quantification of this is shown in Figure 1 SF1C,D and should be moved to the main panel.

We agree that these data are important, but we’re out of space in Figure 1. We therefore think it is best to leave the data in the supplement. Another option would be to split Figure 1 into two figures – we would be happy to do this if requested by the reviewer.

5. p17-18: To support the claim that accumulation of sphingolipid would inhibit the Ypk1/2-dependent increase in Cln3, the authors could monitor the phosphorylation status of Ypk1/2 in sec7 mutants and/or in cells incubated with phytosphingosine. Alternatively, they could analyze whether addition of phytosphingosine on ypk1∆ or ypk1-as ypk2∆ mutant leads to a further decrease of Cln3 level or not.

In a previous study, we showed that addition of PHS to cells causes a rapid loss of TORC2-dependent phosphorylation of Ypk1/2, consistent with a model in which accumulation of PHS inhibits Ypk1/2-dependent signals that promote production of Cln3 (PMID: 29290562). However, we also found that inactivation of Sec7 during growth in G1 phase leads to an increase in TORC2-dependent phosphorylation of Ypk1/2, consistent with a previous study that found similar results for Sec6 in cells synchronized with mating pheromone (PMID: 28794263). We include a discussion of both of these results in the Discussion. To explain the difference in the results, we hypothesize that there are distinct pools of Ypk1/2 at different locations in the cell that undergo differential regulation. One pool of Ypk1/2 could be at the ER, where it controls sphingolipid synthesis and Cln3 levels, and is subject to feedback control by excess sphingolipids. A distinct pool at the plasma membrane could become hyperactive in response to inactivation of Sec7 but not have the capability to influence Cln3 levels. We include a discussion of evidence for two distinct pools of Ypk1/2, as well as evidence for connections between Cln3 regulation and the ER. The complex connections between TORC2, Ypk1/2, sphingolipids, the ER, and Cln3 suggest that we have long ways to go to understand how cell growth and cell cycle progression are mechanistically linked.

Data presentation1. figure 1E: It should be mentioned that it represents the quantification of the WB displayed in D, and if the signal was normalized to a loading control. Please explain why this was not done if it was not.

We have now mentioned that the data in Figure 1E are a quantification of the western blot data in Figure 1D. We wanted the data to provide a measure of the increase in the number of Cln3 molecules per cells as a function of growth. Therefore, we did not normalize to the loading control because levels of the loading control protein increase with growth, so normalizing to the loading control would partially cancel out the growth-dependent increase in Cln3 protein molecules per cell.

2. Figure 3 and figure 3 —figure supplement 1: Two different genotypes are used 2xWHI5 and Wild type + WHI5. It is not clear whether there is a difference between them. If there is none, the use of only 1 genotype would help for the comprehension. It there is a difference, it should be explained.

We agree that it was unclear. The 2xWHI5 genotypes are the same in Figure 3 and Figure 3 —figure supplement 1. The figure inset has been changed to “2xWHI5” to avoid confusion. The strain table also denotes which genotypes are used in each figure panel to help avoid confusion.

3. Figure 3D: Contrary to the authors sentence (p9 3rd paragraph) Cln3 level is not displayed on the panel. If it is relevant, it should be shown. If not, the sentence should be changed. The authors should add a loading control for this panel, which is not shown in figure 3 —figure supplement 1E as stated in the legend of the figure.

The text has been changed to reflect that Cln3 is not in this figure, and a loading control is now included. This figure is now Figure 3 —figure supplement 1C.

4. Figure 5: the panel of the loading control is missing in the figure, while it is mentioned in the legend. It should be added.

We have included a loading control in Figure 5.

5. Figure 6 —figure supplement 1B is not cited in the text. A loading control should be added to the western blot.

Figure 6 —figure supplement 1B is now cited in the text, and we have included the loading control in Figure 6 —figure supplement 1B

6. p12, 4th paragraph; figure 7C and 7D are not cited accurately in the text.

This has been corrected.

7. Table 1: Some strains are not referenced correctly to the corresponding figures. For DK3572, it should be S3C, for DK3743, it should be S3D. The strain 2xWHI5 used in figure 3 —figure supplement 1 is not referenced in the table 1.

The strains listed above were corrected in Table 1 and the other strain figure references were double checked and corrected as necessary.

8. Overall, the manuscript is quite long with some repeats, and would benefit from some shortening.– Figure 4 A,B and the figure 5 —figure supplement 1 could be merged in a unique figure and the corresponding text simplified, as the conclusions drawn from these results are similar.

We tried to do this but found that merging the two figures disrupted the logic in the text describing the figures.

– p12, 3rd paragraph; the first and third sentence are redundant. It should be rephrased.

We agree and have edited accordingly.

– The Discussion could be shortened as results or interpretations already presented in Results are merely rephrased, for instance p15 1st paragraph, p16 4th and 5th paragraphs…

We agree and have edited the section that starts on p. 15. We also deleted the section on Sch9 in the Discussion because it repeated information from the Results section.